# Sensory transduction is required for normal development and maturation of cochlear inner hair cell synapses

John Lee[1,2], Kosuke Kawai[2], Jeffrey R Holt[2,3]*, Gwenaëlle SG Géléoc[2]

[1]Speech and Hearing Bioscience & Technology Program, Division of Medical Sciences, Harvard University, Boston, United States; [2]Department of Otolaryngology, Boston Children's Hospital and Harvard Medical School, Boston, United States; [3]Department of Neurology, Boston Children's Hospital and Harvard Medical School, Boston, United States

**Abstract** Acoustic overexposure and aging can damage auditory synapses in the inner ear by a process known as synaptopathy. These insults may also damage hair bundles and the sensory transduction apparatus in auditory hair cells. However, a connection between sensory transduction and synaptopathy has not been established. To evaluate potential contributions of sensory transduction to synapse formation and development, we assessed inner hair cell synapses in several genetic models of dysfunctional sensory transduction, including mice lacking transmembrane channel-like (*Tmc*) *1*, *Tmc2*, or both, in *Beethoven* mice which carry a dominant *Tmc1* mutation and in *Spinner* mice which carry a recessive mutation in transmembrane inner ear (*Tmie*). Our analyses reveal loss of synapses in the absence of sensory transduction and preservation of synapses in *Tmc1*-null mice following restoration of sensory transduction via *Tmc1* gene therapy. These results provide insight into the requirement of sensory transduction for hair cell synapse development and maturation.

*For correspondence:
jeffrey.holt@childrens.harvard.edu

## Editor's evaluation

Deafness is often caused by a defect in mechanotransduction. Lately it has become clear that synaptopathy, a defect at the first synapse in the auditory pathway, also causes hearing loss. Here the authors show that synapses can be lost following a loss of hair cell mechanotransduction, but that restoration of mechanotransduction can prevent the synaptic loss. These results are important for understanding hearing loss and restoration.

## Introduction

Acoustic overexposure and aging can cause significant loss of synaptic connections between inner hair cells (IHCs) and the afferent fibers of spiral ganglion neurons (SGNs), which function to relay information to the brain via the eighth cranial nerve (*Kujawa and Liberman, 2009*; *Kujawa and Liberman, 2006*; *Sergeyenko et al., 2013*). Loss of IHC-SGN synapses, known as cochlear synaptopathy, is evident following noise exposure producing both temporary threshold shifts (TTS) and permanent threshold shifts (PTS), with rapid loss of up to 40–50% of synapses and gradual degeneration of SGNs even in the absence of hair cell death (*Kujawa and Liberman, 2009*). Aging mice show similar synaptopathy, with a steady loss of synapses and a delayed but similar decrease in the number of SGNs that precedes age-related hair cell loss (*Sergeyenko et al., 2013*).

Damage to stereocilia, the mechanosensory microvilli, is also observed in aging and noise-exposed cochleas. Following PTS-inducing noise exposure, irreversible disarray, fusion, and loss of

IHC stereocilia are visible even in the absence of hair cell death (*Wang et al., 2002*). Similarly, IHC stereocilia in aging mice undergo a number of changes including fusion, elongation, and internalization without obvious hair cell loss (*Bullen et al., 2019*). Stereocilia defects compromise the integrity of the mechanosensory transduction apparatus, resulting in absent or impaired sensory transduction and thus contribute to diminished hearing sensitivity (*Pickles et al., 1987*; *Assad et al., 1991*).

Despite the concurrence of mechanosensory insult and cochlear synaptopathy in noise exposure and aging, impaired sensory transduction has not been implicated as a mechanism contributing to the loss of hair cell synapses. To begin to address the contributions of sensory transduction to the development and maturation of IHC-SGN synapses, we evaluated IHC ribbon synapses across multiple developmental timepoints in five genetic models with disrupted sensory transduction, including mice lacking Tmc1, Tmc2, or both ($Tmc1^{\Delta/\Delta}$, $Tmc2^{\Delta/\Delta}$, $Tmc1^{\Delta/\Delta};Tmc2^{\Delta/\Delta}$), *Beethoven* mice which carry a dominant mutation in *Tmc1* (*Bth*) and *Spinner* mice (*Tmie$^{sr}$*), which carry a mutation in *Tmie*. TMC proteins form the pore of hair cell transduction channels (*Pan et al., 2018*) and TMIE is a necessary component of the hair cell mechanosensory transduction complex (*Zhao et al., 2014*). Mutations in these proteins cause transduction dysfunction and hearing loss in mice and humans (*Kurima et al., 2003*; *Vreugde et al., 2002*; *Kawashima et al., 2011*; *Naz et al., 2002*; *Mitchem et al., 2002*; *Zhao et al., 2014*).

Our analyses reveal an unanticipated and complex relationship between sensory transduction, synaptogenesis, and synaptopathy. In mouse models with genetic disruption of sensory transduction, we find that IHC synapses undergo exuberant synaptogenesis during the first postnatal week and that the number of synapses declines drastically over the following few weeks. We also investigated whether restoration of sensory transduction, using an established *Tmc1* gene therapy strategy capable of targeting nearly 100 % of IHCs (*Lee et al., 2020*; *Wu et al., 2021*), preserved normal synaptic development and maturation. We report that *Tmc1* gene therapy in $Tmc1^{\Delta/\Delta}$ mice preserves synapses in mature mice, and that synapse counts were correlated with recovery of ABR thresholds. Together, these data provide insight into synaptic changes associated with sensory transduction and suggest that dysfunction of sensory transduction may contribute to cochlear synaptopathy.

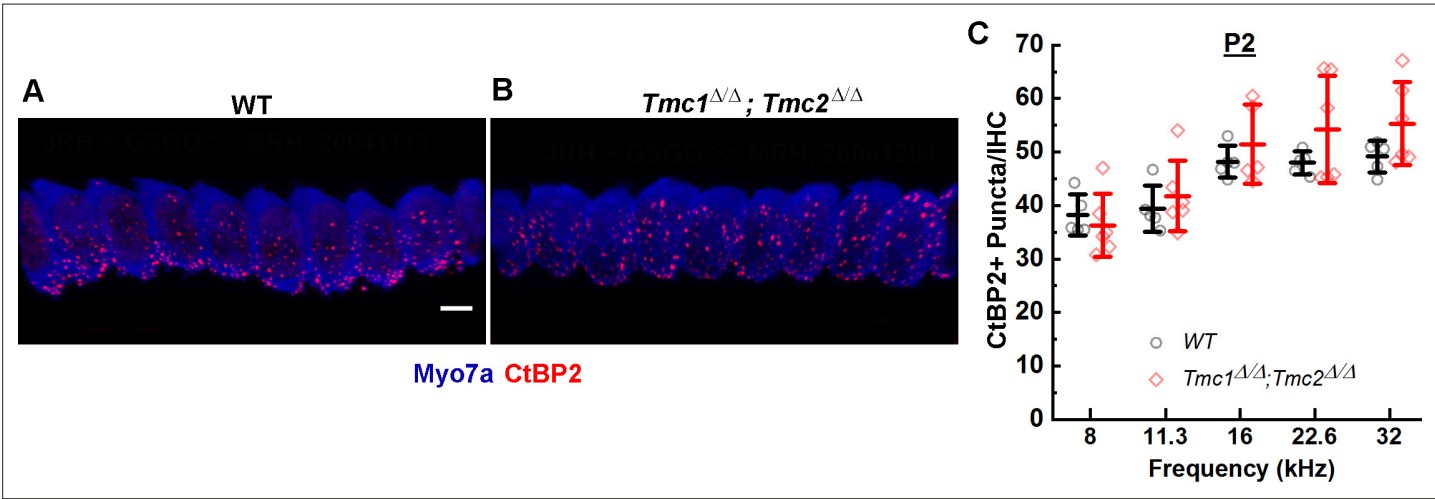

**Figure 1.** CtBP2+ puncta counts in $Tmc1^{\Delta/\Delta};Tmc2^{\Delta/\Delta}$ mice lacking sensory transduction are similar to those in wild-type (WT) mice at postnatal day 2 (P2). (**A**) Representative 3D projections of confocal z-stacks of P2 WT and $Tmc1^{\Delta/\Delta};Tmc2^{\Delta/\Delta}$ (**B**) inner hair cells (IHCs) from the 16 kHz region. Scale bar: 7 µm. The tissue was immunostained with anti-Myosin7a (blue) and anti-CtBP2 (red). (**C**) To determine a mean number of ribbon precursors/IHC, the total number of CtBP2+ puncta from 8 to 10 IHCs was counted for each frequency region. Individual points represent counts from one mouse. Data from WT (black; n = 5) and $Tmc1^{\Delta/\Delta};Tmc2^{\Delta/\Delta}$ (red; n = 5–6) groups are illustrated. Bold lines indicate mean ± SD.

The online version of this article includes the following figure supplement(s) for figure 1:

**Source data 1.** CtBP2+ puncta counts in $Tmc1^{\Delta/\Delta};Tmc2^{\Delta/\Delta}$ mice lacking sensory transduction are similar to those in wild-type (WT) mice at postnatal day 2 (P2).

## Results

### Tmc deletion alters synapse development and maturation

To determine whether initial ribbon formation was influenced by the presence of functional mechanosensory transduction channels, the number of CtBP2-positive puncta per IHC was quantified in five wild-type (WT) and six $Tmc1^{\Delta/\Delta};Tmc2^{\Delta/\Delta}$ mice at postnatal day 2 (P2) from cochlear regions corresponding to 8, 11.3, 16, 22.6, and 32 kHz. Counts of CtBP2-positive puncta were similar in both groups of mice and to those previously reported for WT C57/BL6 mice at P2 (*Huang et al., 2012*). No significant differences in counts were observed between WT and $Tmc1^{\Delta/\Delta};Tmc2^{\Delta/\Delta}$ groups across all frequency regions (*Figure 1*, *Supplementary file 1*), suggesting the initial formation of presynaptic ribbon precursors is independent of the acquisition of sensory transduction, which begins at the base at P0 and progresses tonotopically toward the apex by P4 (*Lelli et al., 2009*).

To assess the consequences of *Tmc1* and *Tmc2* deletion on synaptic development, ribbon synapses were further evaluated at P7, P14, and P28 in WT and $Tmc1^{\Delta/\Delta};Tmc2^{\Delta/\Delta}$ mice. These three timepoints were selected to encompass events crucial to the maturation of the inner ear with P7 following acquisition and maturation of IHC sensory transduction, P14 following hearing onset in mice, and P28

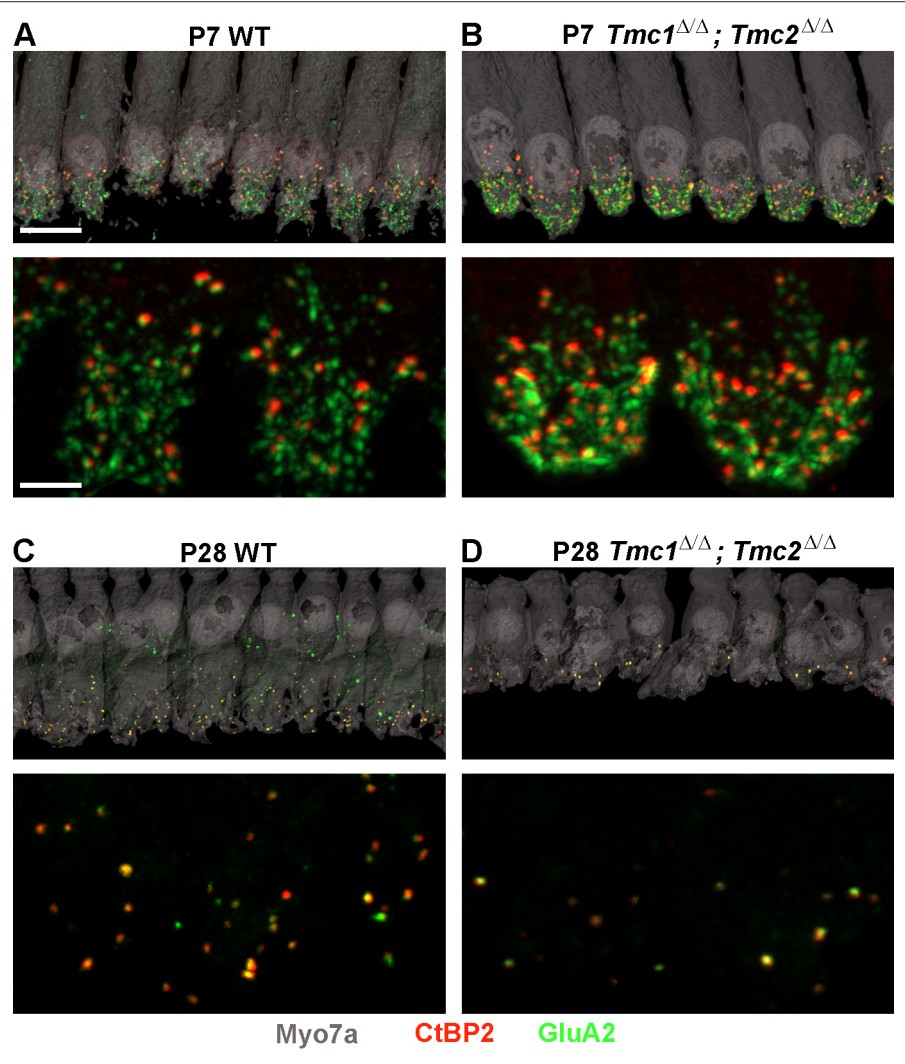

**Figure 2.** Synapse counts are elevated at postnatal day 7 (P7) and reduced at P28 in $Tmc1^{\Delta/\Delta};Tmc2^{\Delta/\Delta}$ mice relative to wild-type (WT) mice. (**A–B**) Representative image of P7 WT and $Tmc1^{\Delta/\Delta};Tmc2^{\Delta/\Delta}$ inner hair cells (IHCs) from 16 kHz region immunostained for anti-Myosin7a (gray), anti-CtBP2 (red), and anti-GluA2 (green). Higher magnification images are shown below. (**C–D**) P28 WT and $Tmc1^{\Delta/\Delta};Tmc2^{\Delta/\Delta}$ IHCs from 16 kHz region. Scale bars: 10 µm (upper) and 5 µm (lower).

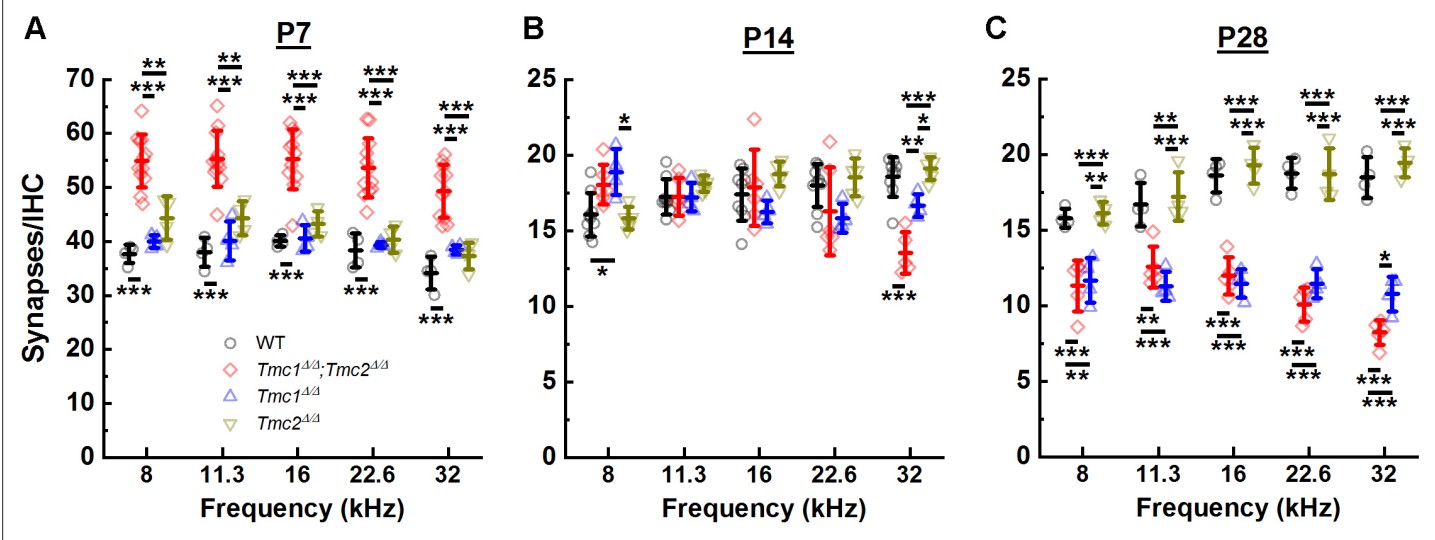

**Figure 3.** Synapse counts were elevated at postnatal day 7 (P7) in the absence of both *Tmc1* and *Tmc2* and diminished at P28 in the absence of *Tmc1* and *Tmc2* or *Tmc1* alone. (**A–C**) The mean number of synapses/inner hair cell (IHC) was calculated for each frequency region. Data from wild-type (WT) (black), *Tmc1^Δ/Δ^;Tmc2^Δ/Δ^* (red), *Tmc1^Δ/Δ^* (blue), and *Tmc2^Δ/Δ^* (dark yellow) groups are shown. Individual points represent counts from one mouse. Temporal changes in synapse counts differed by genotype (two-way ANOVA; p < 0.001 based on two-way interaction between genotype and timepoints for all frequencies; *Supplementary file 3A*). However, genotype-specific trajectories of synaptic development did not vary by frequency (three-way ANOVA; p = 0.73 based on three-way interaction between group, time, frequency; *Supplementary file 3A*). Bolded lines depict mean ± SD. Black horizontal bars and asterisks represent statistically significant differences between group means (multiple pairwise comparisons, *p < 0.05, **p < 0.01, ***p < 0.001; p values are listed in *Supplementary file 2A-C*). Number of cochleas: 4–9 WT, 5–12 *Tmc1^Δ/Δ^;Tmc2^Δ/Δ^*, 4 *Tmc1^Δ/Δ^*, 4–5 *Tmc2^Δ/Δ^*.

The online version of this article includes the following figure supplement(s) for figure 3:

**Source data 1.** Synapse counts were elevated at postnatal day 7 (P7) in the absence of both *Tmc1* and *Tmc2* and diminished at P28 in the absence of *Tmc1* and *Tmc2* or *Tmc1* alone.

reflecting the nearly mature organ of Corti and auditory function. For each timepoint and cochlear region, the average number of synapses per IHC was estimated by counting the total number of presynaptic CtBP2 juxtaposed to postsynaptic GluA2 puncta, divided by the number of IHCs sampled in each region (*Figure 2*).

While ribbon counts were unaltered in P2 *Tmc1^Δ/Δ^;Tmc2^Δ/Δ^* mice, the lack of *Tmc1* and *Tmc2* resulted in elevated synapse counts at P7 across all frequency regions relative to those in WT mice (*Figure 3A*, *Supplementary file 2A*). There were ~43 % more synapses in *Tmc1^Δ/Δ^;Tmc2^Δ/Δ^* mice, suggesting there may be a correlation between the developmental acquisition of sensory transduction in IHCs during the first postnatal week and the developmental increase in the number of ribbons and postsynaptic densities. In WT mice, ~50 % of ribbon synapses are lost between the end of the first and second postnatal weeks (*Sundaresan et al., 2016*). This decrease is thought to result from pruning, refinement, and fusion of ribbons and postsynaptic densities (*Michanski et al., 2019*; *Sundaresan et al., 2016*; *Wong et al., 2014*; *Huang et al., 2012*; *Sendin et al., 2007*). Consistent with these findings, we found a 46–57% reduction in synapses counts in WT mice between P7 and P14 between 8 and 32 kHz. In *Tmc1^Δ/Δ^;Tmc2^Δ/Δ^* mice, a more drastic decrease in synapse counts was evident during the second postnatal week. The number of synapses decreased by 67 to 73% at P14 relative to P7 between 8 and 32 kHz (*Figure 3A and B*). As a result, synapse numbers at P14 did not differ significantly between WT and *Tmc1^Δ/Δ^;Tmc2^Δ/Δ^* groups except at the 32 kHz region where *Tmc1^Δ/Δ^;Tmc2^Δ/Δ^* mice exhibited ~27 % fewer synapse counts (*Figure 3B*, *Supplementary file 2B*).

Following hearing onset in WT mice, ribbon synapse counts remain stable into adulthood (*Michanski et al., 2019*; *Huang et al., 2012*). As expected, our WT synapse counts did not differ significantly between P14 and P28 in any of the frequency regions examined (*Figure 3B and C*). However, synapse counts were markedly decreased in P28 *Tmc1^Δ/Δ^;Tmc2^Δ/Δ^* mice relative to WT mice and relative to P14 *Tmc1^Δ/Δ^;Tmc2^Δ/Δ^* numbers at all frequencies (*Figure 3C*, *Supplementary file 2C*). Though synapse counts remain stable in WT mice post-hearing onset, changes in the distribution of ribbon sizes and continued reduction in the sizes of presynaptic/postsynaptic densities are observed until their full

development around P34 (*Payne et al., 2021*). As with the abnormally elevated synapse counts at P7 in *Tmc1^{Δ/Δ}*;*Tmc2^{Δ/Δ}* mice, the rapid loss of more mature ribbon synapses further suggests a role for sensory transduction in the development of synapses.

## Synapse counts in Tmc single knockouts differ from those of double knockouts

Since *Tmc2* expression coincides with the developmental onset of sensory transduction in OHCs (*Lelli et al., 2009*; *Kawashima et al., 2011*) and IHCs (*Pan et al., 2013*) at P0 in the base and P2-P4 in the apex and is followed several days later by expression of *Tmc1*, we wondered whether genetic deletion of *Tmc2* would cause a delay in the developmental pattern of synapse development. In addition, to determine whether the changes observed in *Tmc1^{Δ/Δ}*;*Tmc2^{Δ/Δ}* synapses were due to deletion of *Tmc1*, we examined synapses in single knockouts of *Tmc1* or *Tmc2*. At P7 and P14, synapse counts in neither *Tmc1^{Δ/Δ}* nor *Tmc2^{Δ/Δ}* mice differed significantly from those of WT mice across all frequencies (*Figure 3A and B*, *Supplementary file 2A and B*), suggesting that expression of either *Tmc1* or *Tmc2* was sufficient for acquisition of normal synapse counts and juxtaposition of presynaptic ribbons and their postsynaptic densities. However, it remains possible that synaptic function and development are altered in mice lacking *Tmc1, Tmc2*, or both in ways that cannot be captured by immunohistochemistry and synapse counts alone.

At P28, there was a significant deviation in the number of synapses observed between *Tmc1^{Δ/Δ}* and *Tmc2^{Δ/Δ}* groups. Synapse counts in *Tmc1^{Δ/Δ}* mice were significantly reduced compared to WT mice and similar to those of *Tmc1^{Δ/Δ}*;*Tmc2^{Δ/Δ}* mice at all frequencies (*Figure 3C*, *Supplementary file 2C*). Synapse counts in *Tmc2^{Δ/Δ}* mice, on the other hand, were not different from those in WT mice at any

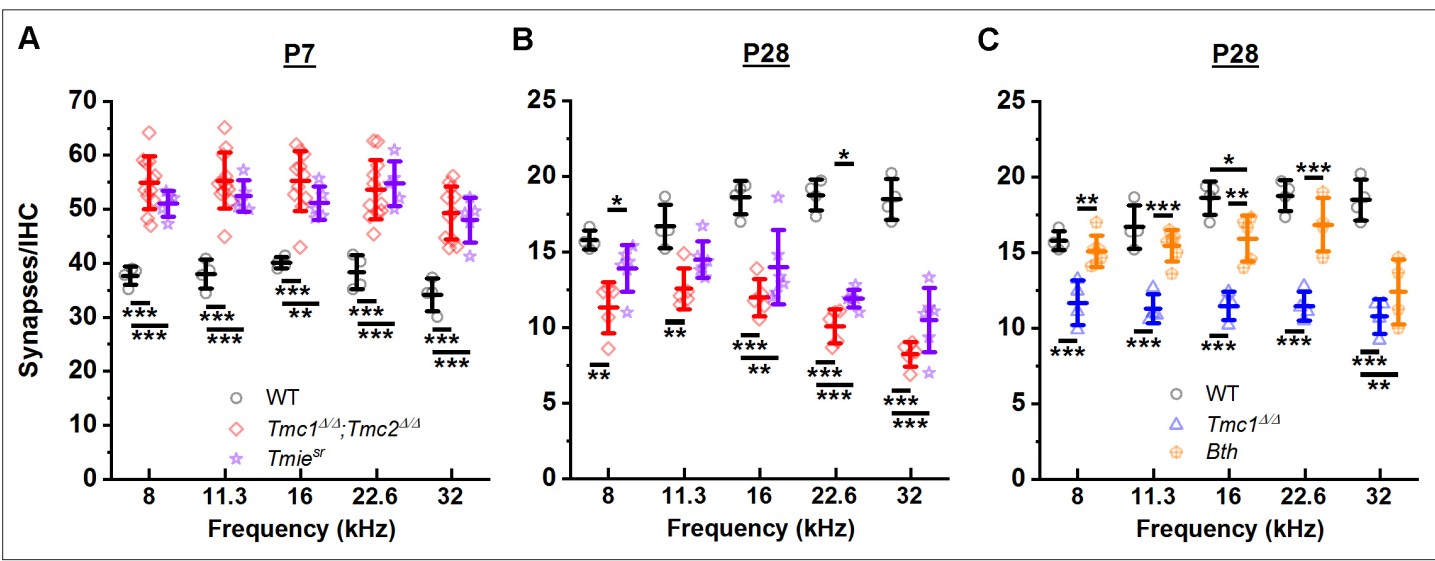

**Figure 4.** Loss of sensory transduction, not of *Tmc1* and *Tmc2* specifically, results in the synaptic differences observed at postnatal day 7 (P7) and P28. (**A–C**) The mean number of synapses/inner hair cell (IHC) at each frequency region. Data from wild-type (WT) (black), *Tmc1^{Δ/Δ}*;*Tmc2^{Δ/Δ}* (red), *Tmc1^{Δ/Δ}* (blue), *Tmie^{sr}* (purple), and *Tmc1^{Bth}* (gold) groups are shown. WT, *Tmc1^{Δ/Δ}*, and *Tmc1^{Δ/Δ}*;*Tmc2^{Δ/Δ}* data are the same as those depicted in *Figure 3*, reprinted here to facilitate comparison. Individual points represent counts from one mouse. In **A** and **B** two-way interactions between genotype and timepoints were statistically significant for all frequencies based on two-way ANOVA (p < 0.001; *Supplementary file 3B*), suggesting the trajectory of synaptic development varies by genotype. The genotype-specific trajectory of synaptic development did not vary by frequency (three-way ANOVA; p = 0.68 based on three-way interaction between genotype, timepoints, and frequency; *Supplementary file 3B*). Frequency-specific synapse counts differed by genotype in **C** (two-way ANOVA; p = 0.002 based on two-way interaction between genotype and frequency; *Supplementary file 3B*). Bolded lines depict mean ± SD. Black horizontal bars and asterisks represent statistically significant differences in group means (multiple pairwise comparisons, *p < 0.05, **p < 0.01, ***p < 0.001; exact p values listed in *Supplementary file 4A-C*). Number of cochleas: 4 WT, 5–12 *Tmc1^{Δ/Δ}*;*Tmc2^{Δ/Δ}*, 4 *Tmc1^{Δ/Δ}*, 5–6 *Tmie^{sr}*, 4–6 *Tmc1^{Bth}*.

The online version of this article includes the following figure supplement(s) for figure 4:

**Source data 1.** Loss of sensory transduction, not of *Tmc1* and *Tmc2* specifically, results in the synaptic differences observed at postnatal day 7 (P7) and P28.

of the frequencies (*Figure 3C*, *Supplementary file 2C*). Thus, it appears the absence of *Tmc1* specifically, in *Tmc1*$^{Δ/Δ}$ and *Tmc1*$^{Δ/Δ}$;*Tmc2*$^{Δ/Δ}$ mice, accounts for the loss of synapses after hearing onset.

## Tmie deletion leads to loss of synapses

While the results obtained from *Tmc1*$^{Δ/Δ}$, *Tmc2*$^{Δ/Δ}$, and *Tmc1*$^{Δ/Δ}$;*Tmc2*$^{Δ/Δ}$ mice implicated a role for sensory transduction in the development and maturation of ribbon synapses, the possibility of ribbon synapses being affected by targeted deletion of *Tmc1*, *Tmc2*, or both via a mechanism unrelated to inhibition of sensory transduction remained a possibility. To determine whether the change in synapse counts was a specific consequence of *Tmc* deletion or a general consequence of the loss of sensory transduction, we investigated deletion of a different gene known to cause loss of hair cell sensory transduction. Synapses were counted in *Spinner* mice, which carry a spontaneous nonsense mutation at the *Tmie* locus (*Tmie*$^{sr}$) (*Mitchem et al., 2002*). TMIE encodes transmembrane inner ear protein, which is an essential component of the transduction channel complex (*Gleason et al., 2009*; *Shen et al., 2008*; *Zhao et al., 2014*). While loss of functional TMIE in homozygous *Spinner* mice causes complete absence of sensory transduction (*Zhao et al., 2014*) and mislocalization of TMC proteins (*Pacentine and Nicolson, 2019*; *Cunningham et al., 2020*), it does not affect TMC expression (*Cunningham et al., 2020*). As shown in *Figure 4A*, *Tmie*$^{sr}$ synapse counts at P7 were significantly elevated compared to WT mice. There were no significant differences between *Tmie*$^{sr}$ and *Tmc1*$^{Δ/Δ}$;*Tmc2*$^{Δ/Δ}$ synapse counts in any frequency region (*Figure 4A*, *Supplementary file 4A*). At P28, *Tmie*$^{sr}$ mice showed decreased synapse counts at all frequencies, with values that did not differ significantly from those in *Tmc1*$^{Δ/Δ}$;*Tmc2*$^{Δ/Δ}$ mice (*Figure 4B*, *Supplementary file 4B*). The similarities in synapse counts in *Tmc1*$^{Δ/Δ}$;*Tmc2*$^{Δ/Δ}$ and *Tmie*$^{sr}$ mice at P7 and P28 provide compelling evidence that the loss of normal synapse development and maturation is a consequence of the absence of sensory transduction and not disruption of a specific gene.

## Altered sensory transduction permeability is inconsequential for synapses

To explore whether the changes in synapses were the result of calcium entry through sensory transduction channels, we performed a similar analysis in *Beethoven* mice with a dominant mutation in *Tmc1*. IHCs in *Tmc1*$^{Bth}$ mice (*Vreugde et al., 2002*) have normal sensory transduction current amplitudes, but reduced calcium permeability relative to control IHCs (*Pan et al., 2013*; *Beurg et al., 2015*; *Corns et al., 2016*). *Marcotti et al., 2006* reported *Tmc1*$^{Bth}$ hair cells have normal resting potentials but altered basolateral currents, leading *Pan et al., 2013* to speculate that altered calcium permeability in *Tmc1*$^{Bth}$ hair cell may have consequences for basolateral hair cell functions, perhaps including synapse development and maturation. No significant differences in P28 synapse counts were evident between *Tmc1*$^{Bth}$ and WT groups except in the 32 kHz region (*Figure 4C*, *Supplementary file 4C*), suggesting reduced calcium entry via sensory transduction channels does not cause a significant loss of ribbon synapses. However, the difference observed in the 32 kHz region suggests a tonotopic role for calcium signaling cannot be discounted.

## Restoration of sensory transduction preserves auditory synapses

Next, we wondered whether gene therapy restoration of sensory transduction could prevent synapse loss. Recent inner ear gene therapy studies have demonstrated the efficacy of a novel utricle injection technique and a synthetic AAV9-PHP.B capsid for transducing nearly 100 % of cochlear hair cells with high specificity (*Lee et al., 2020*) and restoring ABR thresholds with WT *Tmc1* injected into *Tmc1*$^{Δ/Δ}$ mouse inner ears (*Wu et al., 2021*). Although AAV-mediated gene therapy for otoferlin, a synaptic protein, has demonstrated recovery of auditory synapses (*Akil et al., 2019*; *Al-Moyed et al., 2019*), the consequences of *Tmc1* gene therapy on auditory synapses has not been assessed. To characterize the consequences of *Tmc1* gene therapy on IHC synapses, 12 *Tmc1*$^{Δ/Δ}$ mice were injected at P1 with AAV9-PHP.B-*Tmc1* via the utricle and ABRs/synapses were assessed at P28.

At P28, *Tmc1*$^{Δ/Δ}$ mice are profoundly deaf with no measurable ABR thresholds up to 110 dB sound pressure level (SPL) (*Wu et al., 2021*). All 12 *Tmc1*$^{Δ/Δ}$ mice that were injected showed some degree of improvement in auditory function relative to uninjected *Tmc1*$^{Δ/Δ}$ mice (*Figure 5A and B*, *Supplementary file 5*). Recovery was variable across mice, but consistently greater at lower frequencies. On average, the injected mice showed a recovery of 59 dB SPL from 8 to 16 kHz (8 kHz: 61.7 ± 18.3 dB

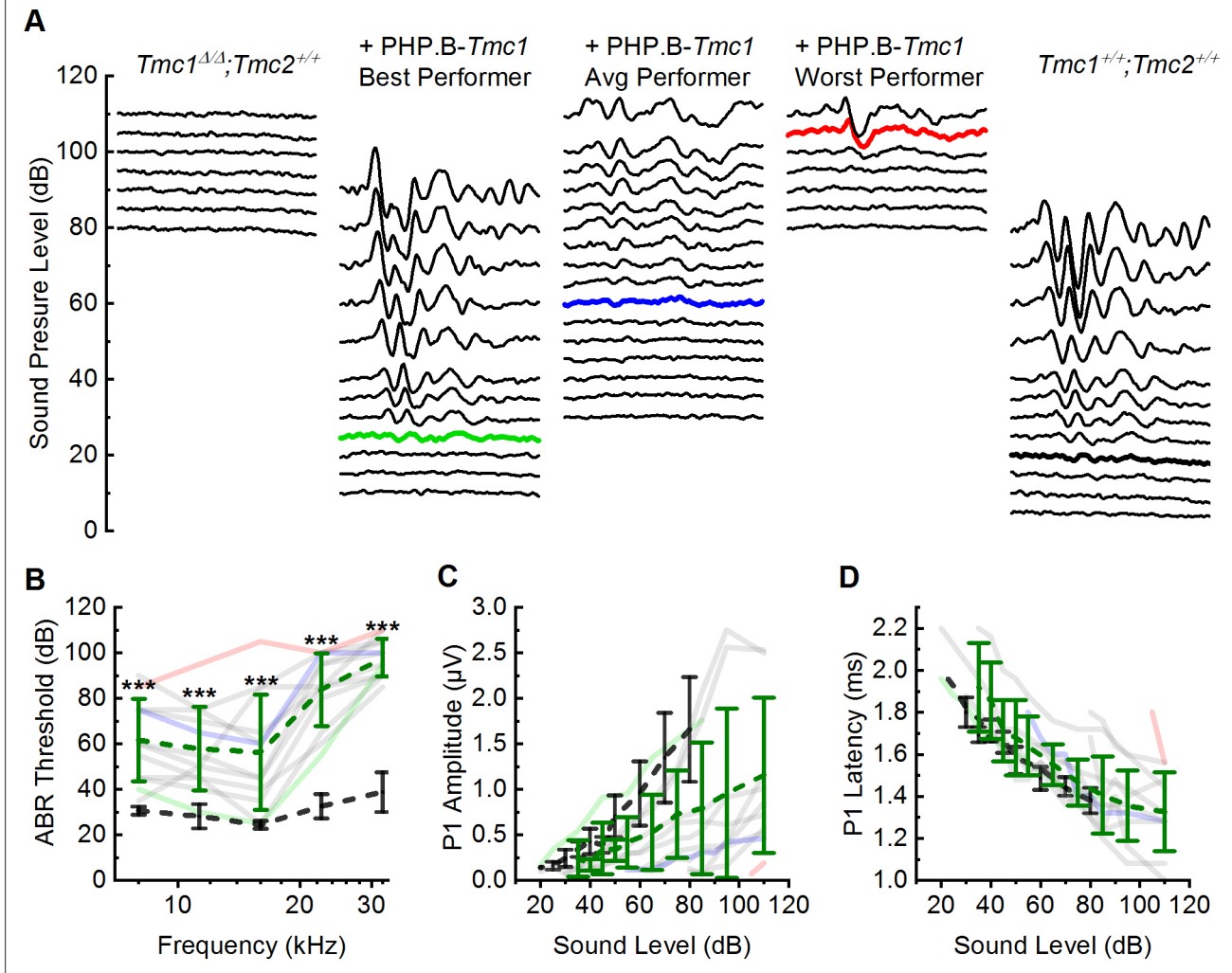

**Figure 5.** AAV9-PHP.B-*Tmc1* restores ABR thresholds in *Tmc1^ΔΔ^* mice. (**A**) Representative ABR waveforms recorded at postnatal day 28 (P28) using 16 kHz tone bursts at sound pressure levels of increasing 5 dB increments. Waveforms from uninjected *Tmc1^ΔΔ^* mouse (left), three *Tmc1^ΔΔ^* mice injected with AAV9-PHP.B-*Tmc1*, representing best (green), average (blue), and worst (red) recovery and one wild-type (WT) control (black) are shown. Thresholds determined by the presence of Peak 1 and indicated by bolded, colored traces. (**B**) ABR thresholds plotted as a function of stimulus frequency for 12 *Tmc1^ΔΔ^* mice injected with AAV9-PHP.B-*Tmc1* tested at P28 (gray traces). Mice with the best (green), median (blue), and worst (red) recovery are indicated and correspond to the best, average, and worst traces in (**A**). Black dotted lines show mean ± SD thresholds from eight previously tested WT mice. Green dotted line shows mean ± SD thresholds from the 12 injected mice. (**C**) Peak 1 amplitudes measured from 16 kHz ABR waveforms (**A**) for 12 *Tmc1^ΔΔ^* mice injected with AAV9-PHP.B-*Tmc1*. Colors correspond to conditions indicated in **B**. (**D**) Peak 1 latencies measured from 16 kHz ABR waveforms (**A**) for 12 *Tmc1^ΔΔ^* mice injected with AAV9-PHP.B-*Tmc1*. Colors correspond to conditions indicated in B.

The online version of this article includes the following figure supplement(s) for figure 5:

**Source data 1.** AAV9-PHP.

SPL; 11 kHz: 57.9 ± 18.4 dB SPL; 16 kHz: 56.3 ± 25.3 dB SPL). The four best performers out of the 12 injected mice demonstrated mean thresholds of 39 dB SPL across 8–16 kHz (*Figure 5B*, *Supplementary file 5*, 8 kHz: 46.3 ± 6.3 dB SPL; 11 kHz: 40.0 ± 7.1 dB SPL; 16 kHz: 31.3 ± 7.5 dB SPL).

Following ABR recording, injected cochleas were immediately harvested and stained for synapses. Average synapse counts were greater in injected *Tmc1^ΔΔ^* mice at all frequency regions (11.3 kHz p = 0.008, 16 kHz p = 0.001, 22.6 kHz p = 0.008, 32 kHz p = 0.012) except at 8 kHz (p = 0.377), than in uninjected *Tmc1^ΔΔ^* mice (*Figure 6A*, *Supplementary file 6A*). Synapse counts were variable across injected mice, similar to ABR threshold recoveries. However, we did note a correlation between the level of ABR threshold recovery and the number of synapses/IHC. We calculated pure tone average (PTA) thresholds from 8 to 22.6 kHz and plotted the values as a function of average number of IHC

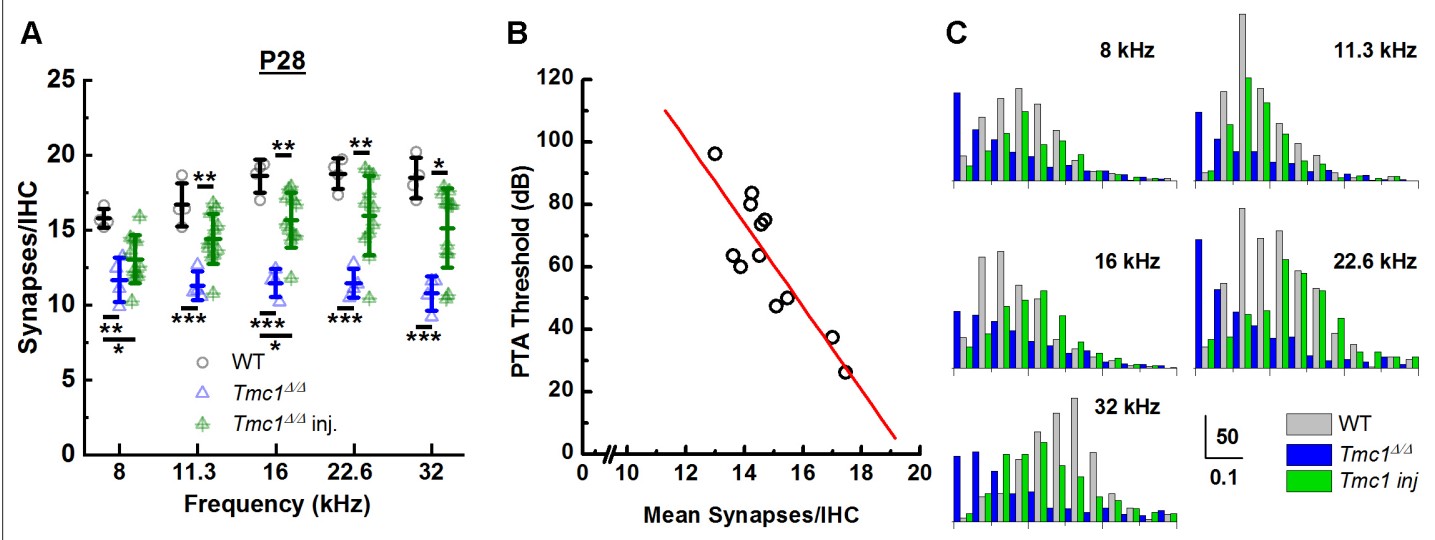

**Figure 6.** AAV9-PHP.B-*Tmc1* preserves synapse counts and ribbon volume distributions in *Tmc1*Δ/Δ mice. (**A**) The mean number of synapses/inner hair cell (IHC) was counted at each frequency region from postnatal day 28 (P28) wild-type (WT) (black), *Tmc1*Δ/Δ (blue), and injected *Tmc1*Δ/Δ (green) mice. WT and *Tmc1*Δ/Δ ribbon counts are the same as those depicted in *Figure 3*. Individual points represent counts from one mouse. Frequency-specific synapse counts did not differ by group (two-way ANOVA; p = 0.49 based on two-way interaction between genotype and frequency; *Supplementary file 3C*). Bolded lines indicate mean ± SD. Black horizontal bars and asterisks represent statistically significant differences in group means (multiple pairwise comparisons, *p < 0.05, **p < 0.01, ***p < 0.001). Number of cochleas: 4 WT, 4 *Tmc1*Δ/Δ cochleas, 11–12 injected *Tmc1*Δ/Δ cochleas. (**B**) Pure tone average (PTA) thresholds were calculated for frequencies between 8 and 22.6 kHz for 12 *Tmc1*Δ/Δ mice injected with 1 μL AAV9-PHP.B-*Tmc1* at P1. PTA thresholds were plotted as a function of the mean synapses/IHC (circles) based on synapse counts from corresponding cochlear regions. The data were fitted with a linear regression (red line) that had a slope of 13 dB/synapse, a correlation coefficient of 0.86; p = 0.00032. (**C**) Histograms showing distributions of ribbon volumes from confocal z-stacks plotted for each frequency region. Scale bars indicate volume counts on the Y-axis and ribbon volumes in μm³ on the X-axis. Data were obtained from four P28 WT (gray) cochleas, four *Tmc1*Δ/Δ (blue) cochleas, and four injected *Tmc1*Δ/Δ (green) cochleas for each of the 8–32 kHz regions.

The online version of this article includes the following figure supplement(s) for figure 6:

**Source data 1.** AAV9-PHP.

synapses across the corresponding cochlear region for 12 mice (*Figure 6B*). The data were fit with a linear regression (*r* = 0.86) which indicated a statistically significant correlation (p = 0.00032) with an improvement of PTA threshold by 13 dB/synapse. The increased synapse counts in *Tmc1*Δ/Δ mice injected with AAV9-PHP.B-*Tmc1* provide a novel line of evidence supporting the potential for *Tmc1* gene therapy in preserving ribbon synapses, a requirement for normal auditory function.

Lastly, we wondered whether the distribution of ribbon volumes was altered by loss and recovery of sensory transduction. Confocal z-stacks of cochleas stained with GluA2 and CtBP2 were acquired using Zeiss Airyscan from four WT mice, four uninjected *Tmc1*Δ/Δ mice, and four injected *Tmc1*Δ/Δ mice with the best ABR threshold recovery and synapse count preservation. For each z-stack, ribbon volumes were calculated using the 'Surfaces' module in Imaris.

The distributions of ribbon volumes are illustrated in *Figure 6C*. Standard deviations of ribbon volumes were compared to evaluate differences in variation between the three groups. P28 uninjected *Tmc1*Δ/Δ mice showed larger distributions of ribbon volumes than WT mice at the same age (*Supplementary file 6B*). These differences were statistically significant at all five frequency regions (*Supplementary file 6C*). Injected *Tmc1*Δ/Δ mice showed narrower distributions more similar to those in WT mice, suggesting *Tmc1* gene therapy preserves both synapse counts and the distribution of ribbon volumes. However, the standard deviations of injected mice were not significantly different from those of uninjected mice (*Supplementary file 6C*).

## Discussion

IHC synapses undergo significant developmental changes during the first few postnatal weeks (*Sobkowicz et al., 1982*; *Huang et al., 2012*; *Yu and Goodrich, 2014*; *Michanski et al., 2019*; *Voorn and*

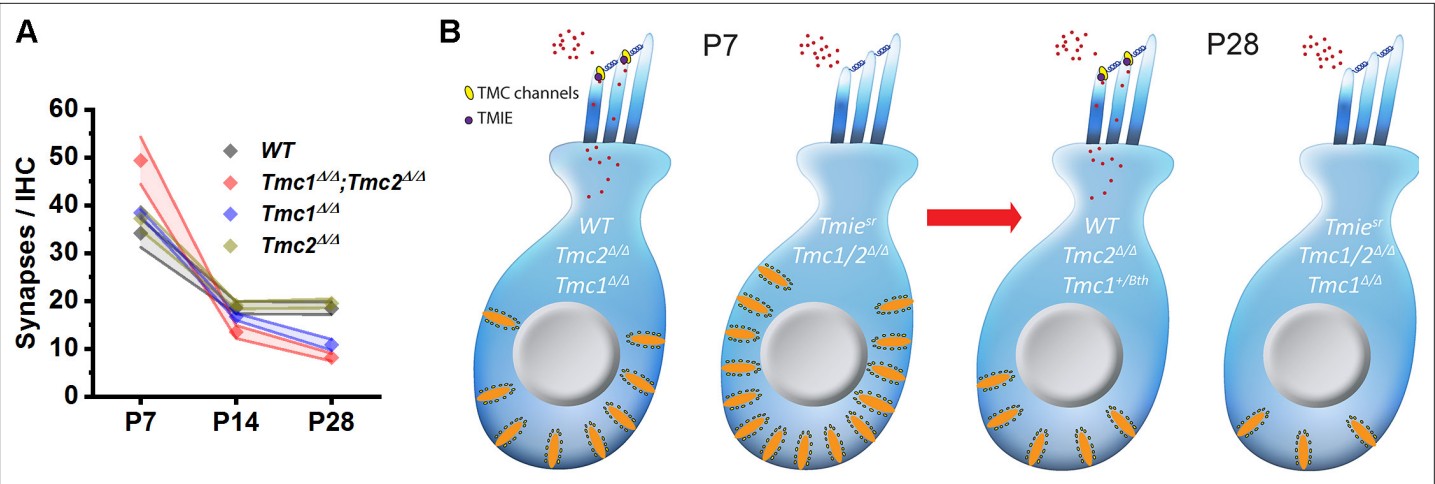

**Figure 7.** Developmental changes in inner hair cell (IHC) synapses in genetic models of sensory transduction dysfunction. (**A**) Number of synapses/IHC from the 32 kHz region of wild-type (WT), $Tmc1^{\Delta/\Delta};Tmc2^{\Delta/\Delta}$, $Tmc1^{\Delta/\Delta}$, and $Tmc2^{\Delta/\Delta}$ cochleas as a function of age. (**B**) Schematic diagram summarizing changes in IHC synapses between postnatal day 7 (P7) and P28 in five genetic models.

*Vogl, 2020*). Sensory transduction is also maturing during this time frame and is required for maturation and maintenance of IHC morphology and a number of biophysical properties (*Corns et al., 2018*). Here, we report that sensory transduction also contributes to the development and maturation of IHC synapses during the first postnatal month.

We began with $Tmc1^{\Delta/\Delta};Tmc2^{\Delta/\Delta}$ mice, which fail to acquire sensory transduction (*Kawashima et al., 2011*), and found they had a similar number of CtBP2-positive puntca to WT mice at P2. This was consistent with prior work suggesting that formation of presynaptic ribbon complexes is an intrinsic property of immature hair cells (*Sobkowicz et al., 1986*). Despite the similarities at P2, synapse counts in $Tmc1^{\Delta/\Delta};Tmc2^{\Delta/\Delta}$ mice were significantly elevated at P7 relative to WT mice (*Figure 7*). These unexpected results are the first to show elevated synapse counts in neonatal mice lacking sensory transduction and suggest that acquisition of sensory transduction may help regulate synaptic pruning and proper synaptic numbers.

In WT mice, the number of synapses progressively increases and then undergoes a reduction during the second postnatal week until the onset of hearing (*Sundaresan et al., 2016*; *Wong et al., 2014*; *Huang et al., 2012*; *Sendin et al., 2007*). While the reduction of synapses still occurred in our $Tmc1^{\Delta/\Delta};Tmc2^{\Delta/\Delta}$ mice, the lack of sensory transduction led to a larger decline in the number of synapses by P14. $Tmc1^{\Delta/\Delta};Tmc2^{\Delta/\Delta}$ mice also displayed reduced synapse counts at P28, in a manner similar to that reported in previous studies assessing synaptic consequences of genetic disruption of synaptic transmission. For example, mice lacking $Ca_v1.3$ $Ca^{2+}$ channels, which are critical for calcium-dependent neurotransmission at IHC synapses (*Platzer et al., 2000*; *Brandt et al., 2005*), lost IHC synapses 4 weeks after birth (*Nemzou N. et al., 2006*; *Brandt et al., 2003*). Another study assessed the synaptic consequences of genetic deletion of vesicular glutamate transporter type 3 (*Kim et al., 2019*) and found roughly half of IHC synapses were lost by 9 weeks of age. Although the genetic targets are different, disruption of synaptic proteins or transduction proteins lead to similar loss of IHC-SGN synapses suggesting they may converge upon similar molecular pathways that regulate synaptic development and maturation.

Genetic deletion of $Tmc1$ or $Tmc2$ alone did not alter synapse counts prior to hearing onset; $Tmc1^{\Delta/\Delta}$ and $Tmc2^{\Delta/\Delta}$ single KO mice exhibited similar counts to those of WT mice at P7. At P28, $Tmc1^{\Delta/\Delta}$ mice synapse counts were reduced like those in $Tmc1^{\Delta/\Delta};Tmc2^{\Delta/\Delta}$ mice, while $Tmc2^{\Delta/\Delta}$ synapses were normal (*Figure 7B*). These differences between $Tmc1^{\Delta/\Delta}$ and $Tmc2^{\Delta/\Delta}$ mice are consistent with a developmental shift from $Tmc2$ to $Tmc1$ mRNA and TMC protein expression that occurs in the cochlea (*Kawashima et al., 2011*; *Kurima et al., 2015*). $Tmc2$ is transiently expressed in the cochlea until P8, while a rise in $Tmc1$ expression is observed between P2 and P22 (*Kawashima et al., 2011*). At P7, synapse counts in $Tmc1^{\Delta/\Delta}$ and $Tmc2^{\Delta/\Delta}$ single KO mice are similar to those of WT mice because both are expressed at that stage and thus available to compensate for the absence of the other. Only in the

absence of both were significantly elevated synapse counts observed at P7. At P28, *Tmc2* is no longer expressed and is thus unavailable to compensate for the absence of *Tmc1* in *Tmc1*$^{\Delta/\Delta}$ mice. Thus, *Tmc1*$^{\Delta/\Delta}$ mice lack sensory transduction, are profoundly deaf, and we find, exhibit similar decreases in synapse counts as *Tmc1*$^{\Delta/\Delta}$;*Tmc2*$^{\Delta/\Delta}$ mice at P28. The absence of native *Tmc2* expression after P8 also explains why *Tmc2*$^{\Delta/\Delta}$ mice are phenotypically normal and have synapse counts similar to WT mice at more mature stages.

To verify the synaptic changes observed at P7 and P28 in the absence of *Tmc1* and *Tmc2* were due to the loss of sensory transduction, synapse counts were also assessed in *Tmie*$^{sr}$ mice and *Tmc1*$^{Bth}$ mice. At both P7 and P28, synapse counts in *Tmie*$^{sr}$ and *Tmc1*$^{\Delta/\Delta}$;*Tmc2*$^{\Delta/\Delta}$ double KO mice were remarkably similar (*Figure 7*), suggesting sensory transduction in general, rather than *Tmc1* or *Tmc2* specifically, helps drive normal development and maturation of IHC synapses. Indeed, while localization of TMC proteins may be disrupted in the absence of TMIE, TMC expression level is not affected (*Cunningham et al., 2020*). Though P28 synapse counts only differed significantly from WT mice at 32 kHz, *Tmc1*$^{Bth}$ synapse counts were not consistently normal. Some mice exhibited counts that were lower than those in WT mice, but greater than those in *Tmc1*$^{\Delta/\Delta}$ mice. The differences in *Tmc1*$^{Bth}$ mice suggest calcium permeability may play a subtle role in regulating synapse numbers. Additional studies assessing the relative contributions of calcium entry via sensory transduction channels will be necessary.

We predict that loss of sensory transduction in *Tmie*$^{sr}$, *Tmc1*$^{\Delta/\Delta}$ and *Tmc1*$^{\Delta/\Delta}$;*Tmc2*$^{\Delta/\Delta}$ mice may lead to hyperpolarization of IHC resting potentials and a lack of depolarizing receptor potentials, leaving voltage-gated Ca$_v$1.3 channels largely deactivated and glutamatergic neurotransmission attenuated. While the hair cell sensory signaling cascade is well established, secondary consequences of disrupted sensory signaling remain obscure. Our data suggest a mechanistic connection between sensory and molecular signaling pathways in the auditory periphery. We propose that sensory signals, in addition to relaying auditory information to the brain, may also converge with molecular signaling pathways that govern development and maturation of IHC-SGN synapses. Disruption of sensory signals, whether genetic in origin or via environmental insults or aging, may disrupt a common pathway leading to synaptopathy and auditory dysfunction. While a number of studies have documented both pathological and normal developmental changes in IHC synapses (*Yu and Goodrich, 2014*; *Voorn and Vogl, 2020*; *Sobkowicz et al., 1982*; *Nouvian et al., 2006*; *Moser et al., 2006*; *Michanski et al., 2019*; *Sergeyenko et al., 2013*; *Fernandez et al., 2020*; *Kujawa and Liberman, 2015*), the molecular mechanisms that regulate development and maturation of IHC synapses and their afferent connections remain unclear.

To further investigate the relationship between sensory transduction and IHC synapses, we discovered that *Tmc1* gene therapy in *Tmc1*$^{\Delta/\Delta}$ mice promoted restoration of sensory transduction, and consequently, preserved synapse counts and ribbon volume distributions. However, variability in both ABR threshold recovery and synapse counts were observed. While the variability in gene therapy recovery is consistent with prior reports (*Nist-Lund et al., 2019*; *Wu et al., 2021*), the source is unclear but may be due to variability in viral injection or viral distribution within the cochlea. Regardless, the variability in the extent of recovery permitted analysis of the relationship between ABR threshold recovery and synapse counts. We found a strong correlation between PTA thresholds and average number of synapses in *Tmc1*$^{\Delta/\Delta}$ mice injected with *Tmc1* gene therapy reagents. Importantly, the data suggest *Tmc1* gene therapy promotes recovery of sensory transduction (*Nist-Lund et al., 2019*) and preservation of synapses, both of which are necessary for normal auditory function. Whether *Tmc1* gene therapy introduced at more mature stages can prevent or promote recovery from synaptopathy remains to be determined.

While our study demonstrates an important role for sensory transduction in development and maturation of IHC-SGN synapses, one caveat is that synapse counts alone are not indicative of the functional status of IHC synapses. Direct assays of synaptic function and synaptic transmission may help validate our results. Verification of normal synapse function, functional sensory transduction, and robust ABR waveforms following inner ear gene therapy may also help inform the continued development of inner ear therapeutics.

This study also raises additional questions regarding the absolute volume of CtBP2-positive ribbons. While our quantification methods allowed for relative comparison of the distribution of ribbon volumes, super resolution microscopy may be better suited to quantify absolute changes in ribbon volumes during development, in the absence of sensory transduction and following gene therapy

recovery of sensory transduction. Furthermore, whether synapses are preferentially lost or recovered on the pillar or modiolar side of IHCs remains to be investigated.

In summary, our study provides novel insight into the role of sensory transduction in synapse development and maturation. While glutamate excitotoxicity has been implicated in the acute loss of synapses observed immediately following noise exposure (*Liberman and Kujawa, 2017*), the mechanisms underlying chronic synaptic changes and neural degeneration of SGNs are unknown. Since noise exposure, aging, and genetic mutations can all cause deficits in sensory transduction, we speculate that damage to the sensory transduction apparatus, regardless of the source of the insult, may affect a common molecular pathway that leads to chronic loss of IHC-SGN synapses and degeneration of SGNs. Identification of these molecular pathways may guide development of future therapeutic strategies that prevent synaptopathy and promote healthy synaptic function.

# Materials and methods

**Key resources table**

| Reagent type (species) or resource | Designation | Source or reference | Identifiers | Additional information |
|---|---|---|---|---|
| Genetic reagent (*Mus musculus*) | C57B/L6-*Cdh23*$^{753A>G}$ | Derived from *Lentz et al., 2010* | C57BL6 | Lentz et al. Dev. Neurobiol (2010) |
| Genetic reagent (*Mus musculus*) | 019146 - B6.129-*Tmc1*$^{tm1.1Ajg}$/J | Available from Jackson Lab, obtained initially from Dr A Griffith (NIH/NIDCD) | Tmc1 Targeted (Reporter, Null/Knockout) | Kurima et al. Nat Genet. (2002) |
| Genetic reagent (*Mus musculus*) | 019147 - B6.129-*Tmc2*$^{tm1.1Ajg}$/J | Available from Jackson Lab, obtained initially from Dr A Griffith (NIH/NIDCD) | Tmc2 Targeted (Reporter, Null/Knockout) | Kawashima et al. J Clin Invest (2011) |
| Genetic reagent (*Mus musculus*) | 003929 - BXA4/Pgn-*Tmie*$^{sr-J}$/J | Available from Jax C57BL/6-*Tmie*$^{sr}$ ('spinner') mice | Spontaneous mutation in Tmie | Stock No. 000543 Mitchem et al. Hum Mol Genet. (2002) |
| Genetic reagent (*Mus musculus*) | *Tmc1*$^{Bth}$ | *Tmc1*$^{Bth/Bth}$ mice were obtained from M Hrabé de Angelis and H Fuchs, Institute of Experimental Genetics, Neuherberg, Germany | Point mutation at residue 412 (M412K) | Vreugde et al. Nat Genet. (2002) |
| Antibody | Anti-CtBP2 (Mouse IgG1 monoclonal) | BD Transduction Laboratories | Cat #: 612044 | Primary antibody, IF (1:200) |
| Antibody | Anti-GluA2 (Mouse IgG2a monoclonal) | Millipore Sigma | Cat #: MABN1189 | Primary antibody, IF (1:2000) |
| Antibody | Anti-MyosinVIIA (Rabbit polyclonal) | Proteus Biosciences | Cat #: 25–6790 | Primary antibody, IF (1:200) |
| Antibody | Anti-Rabbit Alexa Fluor 647 (Donkey polyclonal) | Thermo Fisher Scientific | Cat #: A-31573 | Secondary antibody, IF (1:200) |
| Antibody | Anti-Mouse IgG2a Alexa Fluor 488 (Goat polyclonal) | Thermo Fisher Scientific | Cat #: A-21131 | Secondary antibody, IF (1:1000) |
| Antibody | Anti-Mouse IgG1 Alexa Fluor 546 (Goat polyclonal) | Thermo Fisher Scientific | Cat #: A-21123 | Secondary antibody, IF (1:1000) |
| Other | Vectashield Antifade | Vector Laboratories | Cat #: H-1000–10 | Mounting medium |
| Other | AAV9-PHP.B- CMV-Tmc1e × 1 | *Wu et al., 2021* | | Nucleic acid, Titer: 3.9 E + 13 gc/mL |
| Software, algorithm | Imaris | https://imaris.oxinst.com/ | | |
| Software, algorithm | ImageJ software | http://imagej.nih.gov/ij/ | | |
| Software, algorithm | Eaton-Peabody Laboratories Cochlear Function Test Suite | https://www.masseyeandear.org/research/otolaryngology/eaton-peabody-laboratories/engineering-core | | |

## Mice

WT control mice were C57B/L6 – *Cdh23*$^{753A>G}$ with a corrected *ahl* allele as described by *Lentz et al., 2010*. *Tmc* mutant mice carried mutant alleles of *Tmc1*, *Tmc2*, or both on a C57BL/6 J background (*Tmc1*$^{Δ/Δ}$, *Tmc2*$^{Δ/Δ}$, *Tmc1*$^{Δ/Δ}$;*Tmc2*$^{Δ/Δ}$) (*Vreugde et al., 2002*; *Kawashima et al., 2011*). *Spinner* mice

carrying a spontaneous mutation at the *Tmie* locus (*Tmie$^{sr}$*) on C57BL/6 J backgrounds were obtained from Jackson Laboratories. *Beethoven* mice carrying a dominant mutation in *Tmc1$^{Bth}$* associated with DFNA36 in humans were initially donated by Martin Hrabé de Angelis and Helmut Fuchs at the University of Munich. Genotyping was performed as previously described (*Kawashima et al., 2011*; *Mitchem et al., 2002*). Mice ages P2, P7, P14, and P28 were used for ribbon synapse characterizations. *Tmc1$^{\Delta/\Delta}$* mice ages P0-P1 were used for in vivo delivery of AAV vectors. Mice of both sexes were used in similar numbers and in accordance with protocols approved by the Institutional Animal Care and Use Committee (Protocols #20-02-4149R and #00001240) at Boston Children's Hospital.

## Viral vector preparation

*Tmc1$^{ex1}$* was cloned into an AAV2 vector driven by a CMV promoter and followed with a woodchuck hepatitis virus post-transcriptional regulatory element (WPRE) site, as previously described (*Wu et al., 2021*). The AAV2 vector was then packaged into the AAV9-PHP.B capsid by the Viral Core at Boston Children's Hospital and purified by iodixanol gradient ultracentrifuge followed by ion-exchange chromatography. The titer of genome-containing particles for the AAV2-PHP.B vector was determined using TaqMan quantitative PCR to detect amplicons located in inverted terminal repeats, as previously described (*D'Costa et al., 2016*). The titer of the AAV2/9-PHP.B-CMV-*Tmc1$^{ex1}$* WPRE was calculated to be 3.91E+13 gc/mL. The vector was aliquoted, stored at –80 °C, and thawed immediately before use. Generation and use of AAV vectors were approved by the BCH Institutional Biosafety Committee (Protocol #IBC-P00000447).

## Inner ear injections

Utricle injections were approved by the Institutional Animal Care and Use Committees at BCH (Protocols #20-02-4149R and #00001240) and performed as previously described (*Lee et al., 2020*). Briefly, P1 mice were anesthetized with hypothermia and a postauricular incision was made to expose the semicircular canals. A small puncture into the temporal bone surrounding the utricle was made and a glass micropipette was inserted into the puncture to manually inject 1 µL AAV. After the injection, standard postoperative care was applied.

## ABR acquisition

ABR recordings were performed, as previously described (*Nist-Lund et al., 2019*). Mice were anesthetized with 0.5 mg of ketamine and 0.15 mg of xylazine per 10 g body weight via intraperitoneal injection. Subcutaneous needle electrodes were inserted at the vertex (reference electrode), pinna (active electrode), and rump (ground electrode). Acoustic stimuli were delivered directly into the ear through a custom probe tube speaker/microphone system (EPL PXI Systems) consisting of two electrostatic earphones (CUI Miniature Dynamics) to generate primary tones and a Knowles microphone (Electret Condenser) to record sound pressures from the ear canal. In a sound-proof chamber, mice were presented 5 ms pure tone stimuli of 8, 11.3, 16, 22.6, and 32 kHz at SPL of 10–115 dB in 5 dB increment steps; 512 responses of alternating stimulus polarity were collected and averaged for each SPL. ABR potentials were amplified (10,000 ×), band-pass filtered (0.3–10 kHz), and digitized using custom data acquisition software from the Eaton-Peabody Laboratories Cochlear Function Test Suite.

Waveforms with peak to trough amplitudes greater than 15 µV were discarded by an artifact-reject function. Sound stimuli and electrode voltages were sampled at 40 µs increments using a National Instruments digital input-output board and stored for offline analyses.

## Tissue dissection, immunohistochemistry, and imaging

Temporal bones were dissected and fixed in 4 % paraformaldehyde for 1 hr at room temperature. Temporal bones were then decalcified in 120 mM EDTA for 2 hr for 7-day-old mice and up to 20 hr for 4-week-old mice. Following decalcification, the entire length of the organ of Corti was microdissected in PBS for whole-mount processing. Tissues were then permeabilized by freezing on dry ice in 30 % sucrose and blocked for 1 hr at room temperature in PBS with 0.3 % Triton X + 5 % normal horse serum. Tissues were then stained with the following primary antibodies and incubated at 37 °C overnight: (1) mouse isotype IgG1 anti-C-terminal binding protein 2 (CtBP2, 1:200, BD Transduction Laboratories #612044), (2) mouse isotype IgG2a anti-glutamate receptor 2 (GluA2, 1:2000, Millipore #MABN1189), and (3) rabbit anti-myosin VIIa (Myo7a, 1:200: Proteus Biosciences #25–6790). Tissues

were washed in PBS and incubated for 2 hr at 37 °C with the following secondary antibodies diluted in 1 % normal horse serum + 0.3 % Triton X: (1) goat anti-mouse IgG1 Alexa Fluor 546 (1:1000, Thermo Fisher #A-21123), (2) goat anti-mouse IgG2a Alexa Fluor 488 (1:1000, Thermo Fisher #A-21131), and (3) donkey anti-rabbit Alexa Fluor 647 (1:200, Thermo Fisher #A-31573). Finally, samples were mounted on glass coverslips with Vectashield mounting medium (Vector Laboratories).

Using the 10 × air objective on an LSM 800 (Carl Zeiss), low-power images of the myosin channel were obtained from each microdissected piece. Using a custom ImageJ plugin, a cochlear frequency map was generated by measuring the full length of the cochlea from apex to base using all micro-dissected pieces (*Müller et al., 2005*). Full z-stacks were then acquired at cochlear regions corresponding to five frequencies (8, 11.3, 16, 22.6, 32 kHz) using a 63 × 1.4 NA oil objective lens (Carl Zeiss, z step = 0.36 μm, scaling per pixel: 0.068 μm × 0.068 μm); 9–12 IHCs per field were imaged with z-stacks spanning the entire length of the hair cells. For volume estimations, confocal z-stacks were acquired using the 63 × oil objective with AiryScan processing (Carl Zeiss, z step = 0.18 μm, scaling per pixel: 0.034 μm × 0.034 μm).

## Ribbon synapse counts and volume distribution measurements

Confocal z-stacks were ported to Imaris, an image analysis software, for creation of 3D projections and quantitative analyses of synapse counts and volumes. The 'Spots' module in Imaris was used for automated identification and counting of all ribbons in a given z-stack. All counts were manually reviewed and verified. Synapses were defined as juxtaposition of presynaptic ribbons labeled with anti-CtBP2 with postsynaptic AMPA receptor puncta labeled with anti-GluA2. Juxtaposition was verified manually for every ribbon identified using the 'Spots' module. The total number of synapses was divided by the number of IHCs in the image to calculate the average number of synapses/IHC. For estimation of ribbon volume distributions, confocal z-stacks of ribbon synapses were obtained from P28 control, $Tmc1^{\Delta/\Delta}$, and $Tmc1^{\Delta/\Delta}$ mice injected with AAV2/9-PHP.B-CMV- $Tmc1^{ex1}$WPRE using Airyscan processing and ported to Imaris. 3D projections of Airyscan z-stacks were generated and ribbon volumes were segmented from each projection using the 'Surfaces' module in Imaris. Identical settings were applied across image stacks, including the threshold for 'background subtraction (local contrast)'. This threshold was fixed to ensure digital analyses in Imaris were consistent across images (despite potential variability in immunostaining quality and image acquisition confocal settings) and to prevent subjective biases from affecting ribbon volume calculations. As with the synapse counts, ribbons identified using the 'Surfaces' module were manually verified to be juxtaposed with anti-GluA2 staining. Calculated volumes from each z-stack for each mouse were normalized to the median volume of all ribbons in the z-stack to account for differences in immunostaining quality and image acquisition settings across mice, genotypes, and cochlear regions. Standard deviations of normalized ribbon volumes from each z-stack for each mouse were compared to determine if significant differences in ribbon volume distributions were evident between groups.

## Experimental design and statistical analyses

The Wilcoxon rank sum test was used for comparison of P2 WT and $Tmc1^{\Delta/\Delta};Tmc2^{\Delta/\Delta}$ CtBP2+ puncta counts in *Figure 1C*. For the synapse counts in *Figure 3A-C* and *Figure 4A-C*, three-way ANOVAs were first conducted to evaluate whether genotype differences in the trajectory of synaptic development varied by frequency. In *Figure 3*, the interaction effect between four genotypes (WT, $Tmc1^{\Delta/\Delta};Tmc2^{\Delta/\Delta}$, $Tmc1^{\Delta/\Delta}$, $Tmc2^{\Delta/\Delta}$), three timepoints (P7, P14, P28), and five frequency regions (8, 11.3, 16, 22.6, 32 kHz) was evaluated. In *Figure 4A–B*, the interaction effect between three genotypes (WT, $Tmc1^{\Delta/\Delta};Tmc2^{\Delta/\Delta}$, $Tmie^{sr}$), two timepoints (P7, P28), and five frequency regions (8, 11.3, 16, 22.6, 32 kHz) was evaluated. For *Figure 3-C* and *Figure 4A-B*, two-way ANOVAs were also used to examine the effects of genotype on synapse development at each frequency region. For *Figures 4 and 6*, two-way ANOVAs were used to examine the interaction effect between three genotypes (*Figure 4C*: WT, $Tmc1^{\Delta/\Delta}$, $Tmc1^{Bth}$; *Figure 6A*: WT, $Tmc1^{\Delta/\Delta}$, injected $Tmc1^{\Delta/\Delta}$) and five frequency regions (8, 11.3, 16, 22.6, 32 kHz). Following three-way and two-way ANOVAs, multiple pairwise comparisons were conducted to determine which specific genotype groups differed from one another at each timepoint and frequency region. Six paired comparisons were made between the four groups in *Figure 3A–C* and three paired comparisons were made between the three groups in *Figures 4A–C , and 6A*. The Bonferroni correction was applied to correct for the multiple comparisons and the reported p values

are the original p values multiplied by the number of paired comparisons made. The Wilcoxon rank sum test was also used to compare average ABR thresholds in WT and $Tmc1^{\Delta/\Delta}$ mice injected with $Tmc1$ gene therapy in *Figure 5B*. To evaluate whether the size of ribbon volume distribution differed between control, $Tmc1^{\Delta/\Delta}$ and $Tmc1^{\Delta/\Delta}$ mice injected with $Tmc1$ gene therapy, standard deviations of normalized ribbon volumes from each z-stack were compared using Kruskal-Wallis tests followed by Dunn's multiple comparisons tests using Bonferroni correction. Exact *p* values are reported. Statistical analyses were performed in GraphPad Prism, R, and SAS statistical software. Figures were created using OriginLab, OriginPro.

## Acknowledgements

This work was supported by the NIH/NIDCD grants F32 DC018233 (JL), R01 DC013521 (JRH), and R01 DC008853 (GSGG), Foundation Pour L'Audition, the Jeffrey and Kimberly Barber Fund and the Imaging and Vector Cores at Boston Children's Hospital (BCH IDDRC P30 HD18655). The authors would like to thank Carl Nist-Lund for assistance with injections and ABRs and Stephanie Mauriac and Irina Marcovich for critical review of the manuscript.

## Additional information

### Competing interests

Jeffrey R Holt: holds a patent (62/638,697) on use of AAV9-PHP.B for gene therapy in the inner ear, is a scientific founder of Audition Therapeutics and an advisor to several biotech companies focused on inner ear therapeutics. The authors declare no other conflicts of interest.. The other authors declare that no competing interests exist.

### Funding

| Funder | Grant reference number | Author |
| --- | --- | --- |
| National Institute on Deafness and Other Communication Disorders | RO1 DC013521 | Jeffrey R Holt |
| National Institute on Deafness and Other Communication Disorders | RO1 DC008853 | Gwenaëlle SG Géléoc |
| National Institute on Deafness and Other Communication Disorders | F32 DC018233 | John Lee |

The funders had no role in study design, data collection and interpretation, or the decision to submit the work for publication.

### Author contributions

John Lee, Conceptualization, Formal analysis, Funding acquisition, Investigation, Methodology, Visualization, Writing - original draft; Kosuke Kawai, Formal analysis, Software, Writing – review and editing; Jeffrey R Holt, Conceptualization, Formal analysis, Funding acquisition, Project administration, Supervision, Visualization, Writing – review and editing; Gwenaëlle SG Géléoc, Conceptualization, Funding acquisition, Project administration, Supervision, Writing – review and editing

### Author ORCIDs

Jeffrey R Holt http://orcid.org/0000-0002-7182-8011

### Ethics

This study was performed in strict accordance with the recommendations in the Guide for the Care and Use of Laboratory Animals of the National Institutes of Health. All of the animals were handled according to approved institutional animal care and use committee (IACUC) protocols (#20-02-4149R and #00001240) at Boston Children's Hospital.

Decision letter and Author response
Decision letter https://doi.org/10.7554/eLife.69433.sa1
Author response https://doi.org/10.7554/eLife.69433.sa2

## Additional files

### Supplementary files

• Supplementary file 1. CtBP2+ puncta (mean ± SEM) per inner hair cell (IHC) in postnatal day 2 (P2) wild-type (WT) and $Tmc1^{\Delta/\Delta};Tmc2^{\Delta/\Delta}$ mice.

• Supplementary file 2. Group comparisons of synapse counts at P7 and P14. (A) Group comparisons of postnatal day 7 (P7) synapse counts per inner hair cell (IHC) (mean difference, 95% CI) in wild-type (WT), $Tmc1^{\Delta/\Delta};Tmc2^{\Delta/\Delta}$, $Tmc1^{\Delta/\Delta}$, and $Tmc2^{\Delta/\Delta}$ mice. *p < 0.05, **p < 0.01, ***p < 0.001, non-significant (ns) > 0.05. (B) Group comparisons of postnatal day 14 (P14) synapse counts per inner hair cell (IHC) (mean difference, 95% CI) in wild-type (WT), $Tmc1^{\Delta/\Delta};Tmc2^{\Delta/\Delta}$, $Tmc1^{\Delta/\Delta}$, and $Tmc2^{\Delta/\Delta}$ mice. *p < 0.05, **p < 0.01, ***p < 0.001, non-significant (ns) > 0.05. (C) Group comparisons of postnatal day 28 (P28) synapse counts per inner hair cell (IHC) (mean difference, 95% CI) in wild-type (WT), $Tmc1^{\Delta/\Delta};Tmc2^{\Delta/\Delta}$, $Tmc1^{\Delta/\Delta}$, and $Tmc2^{\Delta/\Delta}$ mice. *p < 0.05, **p < 0.01, ***p < 0.001, non-significant (ns) > 0.05.

• Supplementary file 3. ANOVA analyses as a function of genotype, age and frequency. (A) Three-way and two-way ANOVA analyses of genotypes (wild-type [WT], $Tmc1^{\Delta/\Delta};Tmc2^{\Delta/\Delta}$, $Tmc1^{\Delta/\Delta}$, $Tmc2^{\Delta/\Delta}$), timepoints (P7, P14, P28), and frequencies (8, 11.3, 16, 22.6, 32 kHz) in *Figure 3A–C*. (B) Three-way and two-way ANOVA analyses of genotypes (wild-type [WT], $Tmc1^{\Delta/\Delta};Tmc2^{\Delta/\Delta}$, $Tmc1^{\Delta/\Delta}$, $Tmie^{sr}$, $Tmc1^{Bth}$), timepoints (P7, P28), and frequencies (8, 11.3, 16, 22.6, 32 kHz) in *Figure 4A–C*.(C) Two-way ANOVA analyses of genotypes (wild-type [WT], injected $Tmc1^{\Delta/\Delta}$, $Tmc1^{\Delta/\Delta}$), and frequencies (8, 11.3, 16, 22.6, 32 kHz) in *Figure 6A*.

• Supplementary file 4. Group comparisons at P7 and P28 for WT, $Tmc1^{\Delta/\Delta};Tmc2^{\Delta/\Delta}$, and $Tmie^{sr}$ mice. (A) Group comparisons of postnatal day 7 (P7) synapse counts per inner hair cell (IHC) (mean difference, 95% CI) in wild-type (WT), $Tmc1^{\Delta/\Delta};Tmc2^{\Delta/\Delta}$, and $Tmie^{sr}$ mice. *p < 0.05, **p < 0.01, ***p < 0.001, non-significant (ns) > 0.05. (B) Group comparisons of postnatal day 28 (P28) synapse counts per inner hair cell (IHC) (mean difference, 95% CI) in wild-type (WT), $Tmc1^{\Delta/\Delta};Tmc2^{\Delta/\Delta}$, and $Tmie^{sr}$ mice. *p < 0.05, **p < 0.01, ***p < 0.001, non-significant (ns) > 0.05. (C) Group comparisons of postnatal day 28 (P28) synapse counts per inner hair cell (IHC) (mean difference, 95% CI) in wild-type (WT), $Tmc1^{\Delta/\Delta}$, and $Tmc1^{Bth}$ mice. *p < 0.05, **p < 0.01, ***p < 0.001, non-significant (ns) > 0.05.

• Supplementary file 5. ABR thresholds (mean ± SD) in postnatal day 28 (P28) wild-type (WT) (n = 8) and injected $Tmc1^{\Delta/\Delta}$ (n = 12) mice.

• Supplementary file 6. Group comparisons for synapse counts and synapse volumes at P28. (A) Group comparisons of postnatal day 28 (P28) synapse counts per inner hair cell (IHC) (mean difference, 95% CI) in wild-type (WT), $Tmc1^{\Delta/\Delta}$, and injected $Tmc1^{\Delta/\Delta}$ mice. *p < 0.05, **p < 0.01, ***p < 0.001, non-significant (ns) > 0.05. (B) Normalized distribution of ribbon volumes (median± SD) in postnatal day 28 (P28) wild-type (WT), uninjected $Tmc1^{\Delta/\Delta}$, and injected $Tmc1^{\Delta/\Delta}$ mice. (C) Statistical comparison of standard deviations of normalized ribbon volumes in postnatal day 28 (P28) wild-type (WT), uninjected $Tmc1^{\Delta/\Delta}$, and injected $Tmc1^{\Delta/\Delta}$ mice. See (B) for SD values.

• Transparent reporting form

### Data availability

All data generated or analysed during this study are included in the manuscript and supporting files. Original raw data files have been uploaded to Dryad and are freely available here: https://doi.org/10.5061/dryad.fxpnvx0sb.

The following dataset was generated:

| Author(s) | Year | Dataset title | Dataset URL | Database and Identifier |
|---|---|---|---|---|
| Lee J, Kawai K, Holt J | 2021 | Data from: Sensory transduction is required for normal development and maturation ofcochlear inner hair cell synapses | https://doi.org/10.5061/dryad.fxpnvx0sb | Dryad Digital Repository, 10.5061/dryad.fxpnvx0sb |

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
