## [Editor Report]

Deafness is often caused by a defect in mechanotransduction. Lately it has become clear that synaptopathy, a defect at the first synapse in the auditory pathway, also causes hearing loss. Here the authors show that synapses can be lost following a loss of hair cell mechanotransduction, but that restoration of mechanotransduction can prevent the synaptic loss. These results are important for understanding hearing loss and restoration.

---

## [Decision Letter]

**Decision letter after peer review:**

Thank you for submitting your article "Sensory Transduction is Required for Normal Development and Maintenance of Cochlear Inner Hair Cell Synapses" for consideration by *eLife*. Your article has been reviewed by 3 peer reviewers, and the evaluation has been overseen by a Reviewing Editor and Barbara Shinn-Cunningham as the Senior Editor. The following individuals involved in review of your submission have agreed to reveal their identity: Elisabeth Glowatzki (Reviewer #1); Mark A Rutherford (Reviewer #2); Andreas Neef (Reviewer #3).

All reviewers found that the study is overall well executed and that the results are of significant interest both for auditory neuroscientists as well as potentially for the broader neuroscience community. However, the reviewers also raised a number of criticisms and comments that are listed below.

Figure 6:

There are some issues with panel C. When normalizing to the median, the distribution should be around the value of 1. Clearly, something else was done here but it is not clear what was done. Let's step back for a minute. The reason why Charlie normalizes to the median is two fold – (1) because of the difference in brightness across images, as you stated, and (2) because all of his comparisons are between modiolar and pillar groups in the same images, pooled for different images from the same group. Comparisons between groups are made only in relative terms. Here, you could compare modiolar and pillar and perhaps you should, since the data are already in hand. However, here you are not comparing modiolar and pillar, so, why are you normalizing to the median? The reason to normalize to the median in this case would be to compare the shapes of the distributions, which is basically what you described – a change in the shape where the dKO has a broader distribution. Please mention that you are not detecting potential difference in absolute volume, only differences in the shape of the distribution. Comparing modiolar and pillar would strengthen the paper as well. Another appropriate addition would be to look at GluA2 volumes as well.

The conclusion "suggesting Tmc1 gene therapy preserves both synapse counts and ribbon morphology" should not be based on distributions of normalized ribbon volumes, but absolute ribbon volumes. Is there no better way to address the question? Why is the volume estimate so strongly dependent on the staining? Does Imaris not use a threshold adapted to the dynamic range of the data? What do the raw data look like?

Statistical analysis:

The scarce information about the statistical evaluation is contradictory and unclear:

1. The combination of "assumed to be not normally distributed and "use ANOVA" could be disputed. The assumption of ANOVA is normal distribution, but it may work with samples drawn from distributions violating this assumption.

2. The sentence on line 549 indicates the use of multiple ANOVAs. This probably references the use of one ANOVA for Figure 3 WT vs double mut and one ANOVA across all variants in figure 3 and one ANOVA for figure 4. It would be much clearer if the authors stated that precisely and also made clear that the correction was done for triple testing and also state that the reported p values are one-third the original ones – just to be clear.

3. line 551: "Wilcoxon matched-pairs signed rank test was used for comparison of P2 WT and Tmc1Δ/Δ;Tmc2Δ/Δ ribbon counts".

The source of the "matched" is unclear to me. Obviously, within each WT group and each mutant group, data at different cochlear sections are matched. However, the comparison reported is between WT and mutant across all frequency bands. The data are not matched across the two groups, and therefore it is not clear why the matched-pairs test was warranted.

The statistical description ends as it begins, only that here the word "Gaussian" replaces "normal". Again, it is unclear what is meant by this and why the assumption "Gaussian" is associated with Wilcoxon tests. There is the faint possibility that the authors refer to a specific detail of the implementation of Wilcoxon rank tests. Sometimes, the test uses the assumption of a gaussian distribution of the test statistics to evaluate the probability of the observed rank counts much more quickly. In some programs, this behaviour can be toggled by a parameter switch. Should the authors indeed have this "assumption of Gaussian distribution" in mind, they can safely remove the sentence. For sample sizes of 9 this is never invoked. The speed improvement only kicks in for larger sample sizes. Should they refer to some other assumption, it was not clear.

Given the low number of independent data points, the authors might want to include a statement about the certainty with which they report "no difference". A possibility would be an effect size range they can safely exclude at 90% statistical power.

[Editors' note: further revisions were suggested prior to acceptance, as described below.]

Thank you for resubmitting your work entitled "Sensory Transduction is Required for Normal Development and Maturation of Cochlear Inner Hair Cell Synapses" for further consideration by *eLife*. Your revised article has been evaluated by Barbara Shinn-Cunningham (Senior Editor) and a Reviewing Editor.

The manuscript has been improved but there are some remaining issues that need to be addressed, as outlined below:

Required changes:

1) Figure 6C:

– Maybe it is the log scale that is confusing. Please confirm, for each violin plot, that one-half of the data points are above the median of 1 and one-half of the data points are below the median of 1. How does this look on a linear scale (please show in the response to the reviewers, does not need to be included in the revised manuscript)?

– Figure 6C contains very many individual data points, which cannot be visually separated. Those do not contribute to the reader's information. Almost all detectable information lies in the shape of the violin plots. Those indicate Gaussian shapes for all data groups, but they appear Gaussian on a logarithmic scale. The ordinate values of maximum density are between 0.1 and 0.3 for the KO and around 0.8 to 1 for the WT and rescued KO.

This impression has nothing to do with the actual distribution of the data. I have tried very patiently to reproduce anything resembling those violin plots from the provided data for 8kHz, and I failed.

Just how wrong the impression given by the violin plots is, can readily be appreciated from histograms (of the logarithm of the normalized data). The highest density of data in the blue (KO) cloud is above 1, not below. Interestingly, for the accumulated normalized data BEFORE taking the logarithm, the density is highest below 1, because the distribution of the KO data is very skewed (but not for the WT or KOrescue data).

My recommendation: Please find a program, that correctly reproduces density distributions. If in doubt, whether the algorithms will work with the logarithmic scale, please take the logarithm of your data and then use the violin plot. Then plot on a linear scale and create ticks that reflect the logarithmic scaling.

– It is not per se ok to normalize distributions, then to accumulate the results treat the result as if it represents the original distributions (just with less noise). Only if the samples came from the same distribution (plus a linear scaling), this approach will always yield correct conclusions.

The reason named for normalizing is "variability between stainings". Under some strong assumptions (no offset, i.e. no background staining, and most importantly: a LINEAR relation between staining intensity and detected volume) the data distribution could be meaningfully normalized by division by the median. It is not clear to me that those assumptions have been understood and discussed.

– Raw volume values can be surprisingly small. Zeiss states that the Airyscan can at best resolve 0.14 x 0.14 x 0.35 um. An ellipsoid of these dimensions has a volume of 0.0035um^3. Some of the values in the dataset are smaller. That is counter-intuitive.

– On a positive note, I do not think the discussion about normalization and violin plots is discussion is necessary at all. The unnormalized data (at 8kHz), fully support the following claims about the measured volumes:

mean(Tmc1KO) < mean(WT) < mean(Tmc1rescued)

sd(Tmc1KO) > sd(WT) and no significant difference between the variance of WT and Tmc1rescued.

If I understood the current version correctly, the authors argue that the distributions are different. The statistics of the unnormalized data seems to support this claim.

My advice: work with raw data, accumulate across cochleas, plot histograms rather than violin plots and find a way to capture the vastly different shape of the distributions.

2) In response to a previous reviewer critique the authors say: "The Bonferroni correction was applied following these paired comparisons and the reported p values are 1/6 of the original ones." Let's be careful not to confuse people. For the Bonferroni correction, you need to multiply the original p values, not divide them. In other words, you must divide 0.05 by 3 or 6 to obtain the critical p value to get below to obtain significance. I assume this is a typo, and not a realization that changes the results of the statistical tests. In any case, it should be clarified, and the sentence changed.

3) Reporting of statistical Analysis

This will require some work of the authors or external experts. The standards are pretty clear. *eLife* has supported the push towards more reproducible research and published, among other resources, papers on typical pitfalls and reviews on inadequate reporting:

Meta-Research: Why we need to report more than 'Data were Analyzed by t-tests or ANOVA' Weissgerber et al. 2018 https://doi.org/10.7554/*eLife*.36163.001

Please read those and change the methods section but also the Results section accordingly. The problem is simply that it is not possible to understand what analysis has been performed, even though the data is available, and all variables are clear to me.

Please report the ANOVA design and results.

I am not a statistician, but here is a quick run-down of how ANOVA results are reported – see for instance the *eLife* review cited above.

– Naming the model type (mixed model) – that has been done.

– Naming the dimensionality of the model 3 factors with X levels (4 genotype x 3 Times x 5 Frequency ranges), that's very unclear here.

– Naming the significant effects observed with the model(e.g. effect of time, effect of frequency, effect of genotype, interactions: genotype by time, genotype by frequency), again, nothing is reported about which ones were observed.

By my training, only after the ANOVA shows an effect of factor X, the direction of the effect is checked with t-Tests (as the ANOVA already assumes normality).

What is clear from the numbers in the supplemental data is that 90 separate t-Tests were performed to test for the significance of individual pairs of synapse per IHC counts (5 frequencies x 6 genotype combinations x 3 time points). It is necessary to clarify how the relevant correction factor becomes 6 and not 90.

The authors implicitly report an effect of time at P7 we see this, at P28, we see this. Hence an omnibus ANOVA (freq x genotype x time) would have to show an interaction of time and genotype.

As a side-note: Once more it is unclear how the comparisons are "paired" (line 583). In my reading, 4 values from wt are compared to 4 values from the double KO at the same position of the basilar membrane from animals of the same age. That does not make those comparisons "paired". Truly 'paired' are comparisons of data from the same basilar membrane, i.e. counts at 8, 11.3, 16, … kHz. However, it seems the effect of frequency was not tested (which I presume leads to missing the Tmc2 KO vs wt x Frequency interaction).

I would like to suggest that the authors use less conservative ways of multiple-comparison correction (e.g. Holm). On several occasions (e.g. wt vs Tmc2 KO at P7 8 and 11.3 kHz), this would yield significant differences, where currently there is no effect detected.

Interestingly, this inability of Tmc1 to compensate for the Tmc2 KO at the apex seems to be consistent with the gradient of TMC2  TMC1 replacement reported in Kawashima 2011.

Suggested changes:

– If not done already, suggest to indicate how many images were analyzed per cochlea and how many total synapses are in each group.

– It appears the statistical unit for assessment of significance is # of cochlea, which I would agree is most appropriate. Suggest to state explicitly.

– On line 344: "However, the differences in standard deviations were not statistically significant relative to those uninjected mice" ….. perhaps better: "However, the standard deviations of injected mice were not significantly different from those of uninjected mice"

– Supplementary Table 5b Prism Stats: If you are going to include this, should there not be some description? It seems like you are now comparing the medians. Is this meaningful? Maybe I missed it, but I don't find any description of this in the text. Your response to reviewers agrees it is not meaningful to compare absolute volumes. When it says, "Do the medians vary signif (P<0.05)?" …. are those medians of volumes or medians of SD's? Please include description of Tables. Again, sorry if I just missed it. If the Supplementary files contain irrelevant things, perhaps the irrelevant things should be removed.

– I found some of the supplemental tables difficult to read, like #2 and #3.

– In the Transparent reporting file, what does it mean to say: "Samples were allocated into groups randomly based on genotype and ages."

– Regarding figure 2, I'm not sure I agree that synapse position "could not be assessed due to abnormal IHC morphology." At least, I don't see it in the paper. Maybe I missed it. If IHC morphology is abnormal in these mice, please mention this in the paper if not already. In the images and the schematics, they look normal. I accept that synapse position and morphology are outside the scope of this work.

- Line 587, "divided" should be "multiplied".

---

## [Author Response]

Essential revisions:Figure 6:There are some issues with panel C. When normalizing to the median, the distribution should be around the value of 1.

Normalized ribbon volumes in the original Figure 6C were centered around 100% as percentages of the median. As recommended, Figure 6C has been revised so distributions are normalized with the median value set to 1.

Clearly, something else was done here but it is not clear what was done. Let's step back for a minute. The reason why Charlie normalizes to the median is two fold – (1) because of the difference in brightness across images, as you stated, and (2) because all of his comparisons are between modiolar and pillar groups in the same images, pooled for different images from the same group. Comparisons between groups are made only in relative terms. Here, you could compare modiolar and pillar and perhaps you should, since the data are already in hand. However, here you are not comparing modiolar and pillar, so, why are you normalizing to the median? The reason to normalize to the median in this case would be to compare the shapes of the distributions, which is basically what you described – a change in the shape where the dKO has a broader distribution. Please mention that you are not detecting potential difference in absolute volume, only differences in the shape of the distribution. Comparing modiolar and pillar would strengthen the paper as well. Another appropriate addition would be to look at GluA2 volumes as well.

We feel it is inappropriate to consider immunohistochemistry and confocal microscopy as accurate tools for measuring absolute volumes of synaptic elements, including presynaptic ribbons. There is too much variability in tissue quality, immunostaining quality, etc., which prevents accurate comparison of volumes across samples. However, we do feel that it is accurate to measure volumes within a single image and then normalize for each image. This yields a normalized distribution for each image to itself and allows comparison of volume distributions for each image. We show the normalized data, which we feel provide a valid distribution of CtBP2 volumes that can be compared across samples and experimental conditions.

Comparison of modiolar vs pillar synapses was not part of our study design as we had no reason to hypothesize that manipulation of sensory transduction might have different regional effects within a single cell. Future studies utilizing high resolution techniques methods (i.e. STED, STORM, FIB-SEM) may provide more accurate estimations of ribbon volumes and allow for more precise comparisons.

We have expanded our discussion of these points in the revised manuscript.

The conclusion "suggesting Tmc1 gene therapy preserves both synapse counts and ribbon morphology" should not be based on distributions of normalized ribbon volumes, but absolute ribbon volumes. Is there no better way to address the question? Why is the volume estimate so strongly dependent on the staining? Does Imaris not use a threshold adapted to the dynamic range of the data? What do the raw data look like?

We agree and have revised the text to read “suggesting Tmc1 gene therapy preserves both synapse counts and the distribution of ribbon volumes”. As noted above, we feel super resolution microscopy techniques could be used to allow better estimates of absolute ribbon volumes. Those measurements are beyond the scope of the current study but may be well suited for future studies.

Statistical analysis:The scarce information about the statistical evaluation is contradictory and unclear:1. The combination of "assumed to be not normally distributed and "use ANOVA" could be disputed. The assumption of ANOVA is normal distribution, but it may work with samples drawn from distributions violating this assumption.

Thank you for the comment. We agree the combination of ANOVA and assumption of non-normal distribution are contradictory. The contradictory sentence was removed from the “Experimental Design and Statistical Analyses” paragraph in the methods section. As detailed in our responses below and clarified in the methods section, ANOVA was used for Figures 3A-C, Figures 4A-C, and Figure 6A. We met with Boston Children’s Hospital’s senior biostatistician, Kosuke Kawai (see acknowledgements) and he confirmed it was reasonable to assume normality for our synapse count datasets. However, for CtBP2+ counts (Figure 1B) and ribbon volumes (Figure 6C), we used non-parametric tests because these data may not be normally distributed.

2. The sentence on line 549 indicates the use of multiple ANOVAs. This probably references the use of one ANOVA for Figure 3 WT vs double mut and one ANOVA across all variants in figure 3 and one ANOVA for figure 4. It would be much clearer if the authors stated that precisely and also made clear that the correction was done for triple testing and also state that the reported p values are one-third the original ones – just to be clear.

Thank you for the suggestion. For each panel in Figure 3 (i.e. each timepoint), a total of six paired comparisons was made using ANOVA to analyze the four groups (i.e. A-B, A-C, A-D, B-C, B-D, C-D). The Bonferroni correction was applied following these paired comparisons and the reported p values are 1/6 of the original ones. In Figure 4 and 6A, 3 paired comparisons were made using ANOVA to analyze the three groups (i.e. A-B, A-C, B-C) so reported p values are 1/3 of the original values. We now describe the statistical tests in more detail in the methods section.

3. line 551: "Wilcoxon matched-pairs signed rank test was used for comparison of P2 WT and Tmc1Δ/Δ;Tmc2Δ/Δ ribbon counts".The source of the "matched" is unclear to me. Obviously, within each WT group and each mutant group, data at different cochlear sections are matched. However, the comparison reported is between WT and mutant across all frequency bands. The data are not matched across the two groups, and therefore it is not clear why the matched-pairs test was warranted.

Thank you for the comment. We agree the data are not matched across the two groups and the sentence/statistical test were erroneously included in our description. Statistical analyses of the P2 WT vs. Tmc double KO CtBP2+ puncta counts were repeated using the Wilcoxon rank sum test. P values in Supplemental Table 1 were updated and the sentence was also updated in our “Experimental Design and Statistical Analyses” paragraph. No statistically significant differences between the two groups were evident at any of the frequency regions.

The statistical description ends as it begins, only that here the word "Gaussian" replaces "normal". Again, it is unclear what is meant by this and why the assumption "Gaussian" is associated with Wilcoxon tests. There is the faint possibility that the authors refer to a specific detail of the implementation of Wilcoxon rank tests. Sometimes, the test uses the assumption of a gaussian distribution of the test statistics to evaluate the probability of the observed rank counts much more quickly. In some programs, this behaviour can be toggled by a parameter switch. Should the authors indeed have this "assumption of Gaussian distribution" in mind, they can safely remove the sentence. For sample sizes of 9 this is never invoked. The speed improvement only kicks in for larger sample sizes. Should they refer to some other assumption, it was not clear.

Thank you again for the detailed comments/suggestions regarding our statistical analyses. As with the “assumed to be not normally distributed” and “use ANOVA” comment above, we have removed these contradictory sentences from the revised methods sections to clarify the statistical analyses/descriptions.

Given the low number of independent data points, the authors might want to include a statement about the certainty with which they report "no difference". A possibility would be an effect size range they can safely exclude at 90% statistical power.

We recognize that large standard deviations may mask statistically significant differences in means, especially with low numbers of independent data points. We have edited our supplemental tables to reflect the new/updated statistical analysis and to include additional data (i.e. mean, mean difference, SD, SEM, 95% confidence interval, p values). We carefully examined effect size (mean difference and 95% CI) and believe that any significant differences between groups were captured in the statistical analyses.

[Editors' note: further revisions were suggested prior to acceptance, as described below.]

Essential revisions:1) Figure 6C:– Maybe it is the log scale that is confusing. Please confirm, for each violin plot, that one-half of the data points are above the median of 1 and one-half of the data points are below the median of 1. How does this look on a linear scale (please show in the response to the reviewers, does not need to be included in the revised manuscript)?– Figure 6C contains very many individual data points, which cannot be visually separated. Those do not contribute to the reader's information. Almost all detectable information lies in the shape of the violin plots. Those indicate Gaussian shapes for all data groups, but they appear Gaussian on a logarithmic scale. The ordinate values of maximum density are between 0.1 and 0.3 for the KO and around 0.8 to 1 for the WT and rescued KO.This impression has nothing to do with the actual distribution of the data. I have tried very patiently to reproduce anything resembling those violin plots from the provided data for 8kHz, and I failed.Just how wrong the impression given by the violin plots is, can readily be appreciated from histograms (of the logarithm of the normalized data). The highest density of data in the blue (KO) cloud is above 1, not below. Interestingly, for the accumulated normalized data BEFORE taking the logarithm, the density is highest below 1, because the distribution of the KO data is very skewed (but not for the WT or KOrescue data).My recommendation: Please find a program, that correctly reproduces density distributions. If in doubt, whether the algorithms will work with the logarithmic scale, please take the logarithm of your data and then use the violin plot. Then plot on a linear scale and create ticks that reflect the logarithmic scaling.– It is not per se ok to normalize distributions, then to accumulate the results treat the result as if it represents the original distributions (just with less noise). Only if the samples came from the same distribution (plus a linear scaling), this approach will always yield correct conclusions.The reason named for normalizing is "variability between stainings". Under some strong assumptions (no offset, i.e. no background staining, and most importantly: a LINEAR relation between staining intensity and detected volume) the data distribution could be meaningfully normalized by division by the median. It is not clear to me that those assumptions have been understood and discussed.– Raw volume values can be surprisingly small. Zeiss states that the Airyscan can at best resolve 0.14 x 0.14 x 0.35 um. An ellipsoid of these dimensions has a volume of 0.0035um^3. Some of the values in the dataset are smaller. That is counter-intuitive.– On a positive note, I do not think the discussion about normalization and violin plots is discussion is necessary at all. The unnormalized data (at 8kHz), fully support the following claims about the measured volumes:mean(Tmc1KO) < mean(WT) < mean(Tmc1rescued)sd(Tmc1KO) > sd(WT) and no significant difference between the variance of WT and Tmc1rescued.If I understood the current version correctly, the authors argue that the distributions are different. The statistics of the unnormalized data seems to support this claim.My advice: work with raw data, accumulate across cochleas, plot histograms rather than violin plots and find a way to capture the vastly different shape of the distributions.

We appreciate the editor’s attention to this issue. Indeed, we also struggled with how to best represent the data before we settled on the violin plots. To address the editors concerns we replotted the data 24 different ways. In the end, we have followed the editor’s advice and plot the raw data in five histograms, one for each frequency region of the cochlea. We agree this presentation provides the most accurate visual representation of the data for the three conditions, WT, Tmc1 KO and recovery with Tmc1 gene therapy.

2) In response to a previous reviewer critique the authors say: "The Bonferroni correction was applied following these paired comparisons and the reported p values are 1/6 of the original ones." Let's be careful not to confuse people. For the Bonferroni correction, you need to multiply the original p values, not divide them. In other words, you must divide 0.05 by 3 or 6 to obtain the critical p value to get below to obtain significance. I assume this is a typo, and not a realization that changes the results of the statistical tests. In any case, it should be clarified, and the sentence changed.

This was a mistake as noted below in “Suggested changes”. The error was corrected and p values were verified in Prism, R, and SAS. Results of statistical tests were unaltered.

3) Reporting of statistical AnalysisThis will require some work of the authors or external experts. The standards are pretty clear. eLife has supported the push towards more reproducible research and published, among other resources, papers on typical pitfalls and reviews on inadequate reporting:Meta-Research: Why we need to report more than 'Data were Analyzed by t-tests or ANOVA' Weissgerber et al. 2018 https://doi.org/10.7554/eLife.36163.001Please read those and change the methods section but also the Results section accordingly. The problem is simply that it is not possible to understand what analysis has been performed, even though the data is available, and all variables are clear to me.Please report the ANOVA design and results.I am not a statistician, but here is a quick run-down of how ANOVA results are reported – see for instance the eLife review cited above.– Naming the model type (mixed model) – that has been done.– Naming the dimensionality of the model 3 factors with X levels (4 genotype x 3 Times x 5 Frequency ranges), that’s very unclear here.– Naming the significant effects observed with the model(e.g. effect of time, effect of frequency, effect of genotype, interactions: genotype by time, genotype by frequency), again, nothing is reported about which ones were observed.By my training, only after the ANOVA shows an effect of factor X, the direction of the effect is checked with t-Tests (as the ANOVA already assumes normality).What is clear from the numbers in the supplemental data is that 90 separate t-Tests were performed to test for the significance of individual pairs of synapse per IHC counts (5 frequencies x 6 genotype combinations x 3 time points). It is necessary to clarify how the relevant correction factor becomes 6 and not 90.The authors implicitly report an effect of time at P7 we see this, at P28, we see this. Hence an omnibus ANOVA (freq x genotype x time) would have to show an interaction of time and genotype.

The Methods section has been rewritten to describe and report statistical analyses done more accurately. Additional supplemental tables (Supplementary file 6A-6C created to highlight findings of three-way and two-way ANOVAs). A statistician at Boston Children’s Hospital, Kosuke Kawai, was consulted and is now included as a co-author. In the revised manuscript we have rewritten the Methods section and verified the statistical analyses using R and SAS software.

As a side-note: Once more it is unclear how the comparisons are "paired" (line 583). In my reading, 4 values from wt are compared to 4 values from the double KO at the same position of the basilar membrane from animals of the same age. That does not make those comparisons "paired". Truly 'paired' are comparisons of data from the same basilar membrane, i.e. counts at 8, 11.3, 16, … kHz. However, it seems the effect of frequency was not tested (which I presume leads to missing the Tmc2 KO vs wt x Frequency interaction).

As noted above, the methods section was rewritten to clarify the statistical analyses done on our datasets.

I would like to suggest that the authors use less conservative ways of multiple-comparison correction (e.g. Holm). On several occasions (e.g. wt vs Tmc2 KO at P7 8 and 11.3 kHz), this would yield significant differences, where currently there is no effect detected.Interestingly, this inability of Tmc1 to compensate for the Tmc2 KO at the apex seems to be consistent with the gradient of TMC2  TMC1 replacement reported in Kawashima 2011.

We decided to stick with the Bonferroni correction and not modify correction to be less conservative.

Suggested changes:– If not done already, suggest to indicate how many images were analyzed per cochlea and how many total synapses are in each group.

Done*.*

– It appears the statistical unit for assessment of significance is # of cochlea, which I would agree is most appropriate. Suggest to state explicitly.

This is now stated in the figure legends.

– On line 344: "However, the differences in standard deviations were not statistically significant relative to those uninjected mice" ….. perhaps better: "However, the standard deviations of injected mice were not significantly different from those of uninjected mice"

Sentence was changed per editors’ suggestion.

– Supplementary Table 5b Prism Stats: If you are going to include this, should there not be some description? It seems like you are now comparing the medians. Is this meaningful? Maybe I missed it, but I don't find any description of this in the text. Your response to reviewers agrees it is not meaningful to compare absolute volumes. When it says, "Do the medians vary signif (P<0.05)?" …. are those medians of volumes or medians of SD's? Please include description of Tables. Again, sorry if I just missed it. If the Supplementary files contain irrelevant things, perhaps the irrelevant things should be removed.

Thank you for the suggestion. We agree the Supplementary Table 5 Prism Stats tab was contained irrelevant data and the tab was removed. The “Do the medians vary signif” referred to the medians of SDs. The updated Supplementary file 5 now contains three tabs, each corresponding to a supplemental table.

– I found some of the supplemental tables difficult to read, like #2 and #3.

Supplemental Tables 2 and 3 were re-structured to be easier to read. Each table was broken down into sub tables corresponding to their respective figures (i.e. Supplementary file 2A-C Figure 3A-C; Supplementary file 3A-C Figure 4A-C). Supplementary file legends were updated to reflect these new tables.

– In the Transparent reporting file, what does it mean to say: "Samples were allocated into groups randomly based on genotype and ages."

There was no randomization. We state “Samples were allocated into groups based on genotypes and ages”. The word random does not appear. Groups contained mice of the same genotype and age. Cochlea samples from WT, P7 mice were grouped together while cochleas from Tmc1 KO, p7 were placed in a separate group. i.e. groups were based on genotype and age.

– Regarding figure 2, I'm not sure I agree that synapse position "could not be assessed due to abnormal IHC morphology." At least, I don't see it in the paper. Maybe I missed it. If IHC morphology is abnormal in these mice, please mention this in the paper if not already. In the images and the schematics, they look normal. I accept that synapse position and morphology are outside the scope of this work.

We have not quantified that IHC morphology. At this point it is just an impression. In the absence of data, we’d rather not go on record stating it is abnormal. We agree, IHC morphology and synapse position are beyond the scope of the current manuscript and are better left for a future publication.

- Line 587, "divided" should be "multiplied".

Thank you for noting this typo. The sentence has been corrected.